# The architecture and operating mechanism of a cnidarian stinging organelle

Ahmet Karabulut [1]✉, Melainia McClain [1], Boris Rubinstein[1], Keith Z. Sabin[1], Sean A. McKinney[1] & Matthew C. Gibson [1,2]✉

The stinging organelles of jellyfish, sea anemones, and other cnidarians, known as nemato-cysts, are remarkable cellular weapons used for both predation and defense. Nematocysts consist of a pressurized capsule containing a coiled harpoon-like thread. These structures are in turn built within specialized cells known as nematocytes. When triggered, the capsule explosively discharges, ejecting the coiled thread which punctures the target and rapidly elongates by turning inside out in a process called eversion. Due to the structural complexity of the thread and the extreme speed of discharge, the precise mechanics of nematocyst firing have remained elusive[7]. Here, using a combination of live and super-resolution imaging, 3D electron microscopy, and genetic perturbations, we define the step-by-step sequence of nematocyst operation in the model sea anemone *Nematostella vectensis*. This analysis reveals the complex biomechanical transformations underpinning the operating mechanism of nematocysts, one of nature's most exquisite biological micro-machines. Further, this study will provide insight into the form and function of related cnidarian organelles and serve as a template for the design of bioinspired microdevices.

[1] Stowers Institute for Medical Research, Kansas City, MO, USA. [2] Department of Anatomy and Cell Biology, The University of Kansas School of Medicine, Kansas City, KS, USA. ✉email: ahk@stowers.org; MG2@stowers.org

Cnidarian nematocysts are complex subcellular weapons with highly specialized forms and functions[1,2]. Nematocysts are Golgi-derived intracellular organelles comprised of venomous threads enclosed within a pressurized capsule[3,4]. When triggered, the capsule discharges, ejecting its thread as a harpoon that penetrates targets, delivering a cocktail of neurotoxins[5–10]. At the cellular level, nematocyst discharge is among the fastest mechanical processes in nature, known to be completed within 3 milliseconds in *Hydra* nematocysts[11,12]. Measurements performed on high-speed video of *Hydra* stenoteles reveal that the initial phase of pressure-driven capsule explosion and subsequent thread ejection occurs in as fast as 700 nanoseconds[12]. This initial stage of explosive discharge is comparable to other ultra-fast projectile systems found in nature such as fungal spore discharge, pollen ejection, and discharge of the ballistic organelles of dinoflagellates[13,14].

Previous studies indicate that the high-speed of nematocyst discharge is driven by the accumulation of osmotic pressure inside the capsule by a matrix of cation binding poly-γ-glutamate polymers (PGs) and the elastically stretched capsule wall releasing energy by a powerful spring-like mechanism during discharge[2,12,15,16]. Upon triggering, but prior to discharge, the capsule approximately doubles in volume due to the rapid influx of water[17]. This causes the matrix to swell osmotically and stretches the capsule wall[2,18]. This energy is subsequently utilized to eject the thread with high velocity, which impacts and penetrates target tissue. The later phases of nematocyst discharge involve the elongation of the thread, which proceeds on a slower timescale and is completed in milliseconds[11]. During this phase, the nematocyst thread undergoes a shape transformation, turning inside-out through a process called eversion which is caused by the release of both osmotically generated pressure and elastic energy stored in the thread[17,19,20]. Thus, the nematocyst operates in distinct phases that involve an initial phase of piercing the target and later phases of eversion to form a lumen.

Nematocyst characteristics vary significantly among different cnidarian species, exhibiting diversity in capsule size and thread morphology, but all retain a similar mechanism of operation involving an evertible tubule driven by explosive ejection[2,21–23]. To explore nematocyst biology in a genetically tractable system, here we interrogate the operation of the nematocyst thread in the sea anemone *Nematostella vectensis*. *Nematostella* harbors two types of nematocysts: microbasic p-mastigophores and basitrichous isorhizas, the latter having short and long varieties[24,25]. In sea anemones, nematocyst capsules are sealed by three apical flaps connected to the stinging thread[26–28]. This thread is composed of two distinct sub-structures: a short, rigid, and fibrous shaft and a long thin tubule decorated with barbs[17,22]. The shaft is composed of three helically coiled filaments, and is initially ejected as a compressed projectile, piercing the target, and later everts to form a lumen through which the remainder of the thread, the tubule, is released[17]. While it is known that shaft eversion entails a geometric transformation from a tightly compressed coil to a hollow syringe, the mechanisms driving this process are poorly understood. Further, tubule eversion significantly differs from that of the triple helical shaft, as the tubule everts by turning inside-out in the absence of helical filaments[26,29]. The release of pressure and elastic energy stored in the capsule is theoretically sufficient to drive the initial ejection and penetration of the shaft, however, additional energy sources are likely to be required for further elongation of the thread[5,19,20]. Due to the speed and complexity of these events, the precise stages of discharge and eversion have thus far remained elusive.

Here, we demonstrate the structural composition and mechanical transformations of both the shaft and the tubule during distinct phases of nematocyst discharge in *Nematostella*, and further report the operating mechanism of the nematocyst thread sub-structures. Our analysis reveals the complex structure and the sophisticated biomechanical transformations underpinning the operational mechanism of nematocysts.

## Results

**Visualizing nematocysts in *Nematostella*.** To understand the distribution of stinging cells (nematocytes) and their nematocysts in *Nematostella*, we first created a transgenic line expressing EGFP in nematocytes under the control of the nematogalectin promoter region (nematogalectin > *EGFP*; Fig. 1a). Nematogalectin is a major component of the nematocyst, and it is incorporated into the thread structure during its morphogenesis[30]. This protein is thought to act as a substrate for the assembly of other structural proteins into the thread, thus its temporal expression defines a useful window for visualizing nematocytes[30]. Live imaging of transgenic primary polyps showed that the tentacles were heavily populated with EGFP+ nematocytes bearing the long form basitrichs (Fig. 1a^I). The body column was populated with the shorter variety along with a few p-mastigophores. Intriguingly, we found that nematocytes were connected through neurite-like processes which formed local networks (Fig. 1a^II, arrow). Nematocytes are known to form synapses and act as afferents or effectors but can also operate cell-autonomously[31–34]. Thus, the observed networks might function in regulating collective behavior and coordinated activity of nematocyte populations[35]. In EGFP+ nematocytes, fluorescence was detected throughout the cytoplasm and the sensory apparatus but was excluded from the capsule (Fig. 1b). The capsule wall and thread are built, in part, of minicollagens which allow the construction of a variety of structural fibers by cross-linking[36–41]. We exploited this to visualize the capsule content by treating live animals with fluorescent TRITC which was incorporated into the nematocyst thread during its maturation, presumably through a reaction with minicollagens[42,43].

In basitrich type nematocysts, TRITC incorporation was seen only after thread invagination (Supplementary Fig. 1a, arrows). In contrast, nematocytes harboring maturing threads were devoid of TRITC (Supplementary Fig. 1a, dashed arrows). This suggests that the dye specifically accumulates in invaginated parts of the thread inside the capsule. We found that shRNA-mediated knockdown of the *nematogalectin*-like gene *Nemve1_232014* resulted in abnormal capsules and prevented thread formation (Supplementary Fig. 1b). This suggests that lectins play a critical role in thread and capsule morphogenesis (Supplementary Fig. 1b, dashed arrows). We further confirmed that knockdown resulted in a two-fold reduction of *Nemve1_232014* mRNA expression by qPCR, suggesting that this protein must be present in abundance for proper assembly of the thread and capsule (Supplementary Fig. 1c). Utilizing a combination of nematogalectin>*EGFP* and TRITC dye labeling, we next analyzed the architecture of the thread from its development to its final morphology after firing (Fig. 1b, c; Supplementary Movie 1). In contrast to the dense shaft of p-mastigophores (Fig. 1c, arrows), in which the dye intensity was very high compared to the tubule, fluorescent TRITC incorporated with similar intensity in both the shaft and tubule of basitrichs (Fig. 1c, dashed arrows). The more uniform labeling of basitrichs and their prevalence in primary polyps led us to investigate thread operation in this nematocyst type.

**The architecture of undischarged nematocysts.** To analyze the structure of the shaft and tubule and thereby determine their functionality, we next performed 3D-reconstruction of undischarged basitrich capsules from serial sections using scanning electron microscopy (SEM; Fig. 1d; Supplemental Movie 2). We

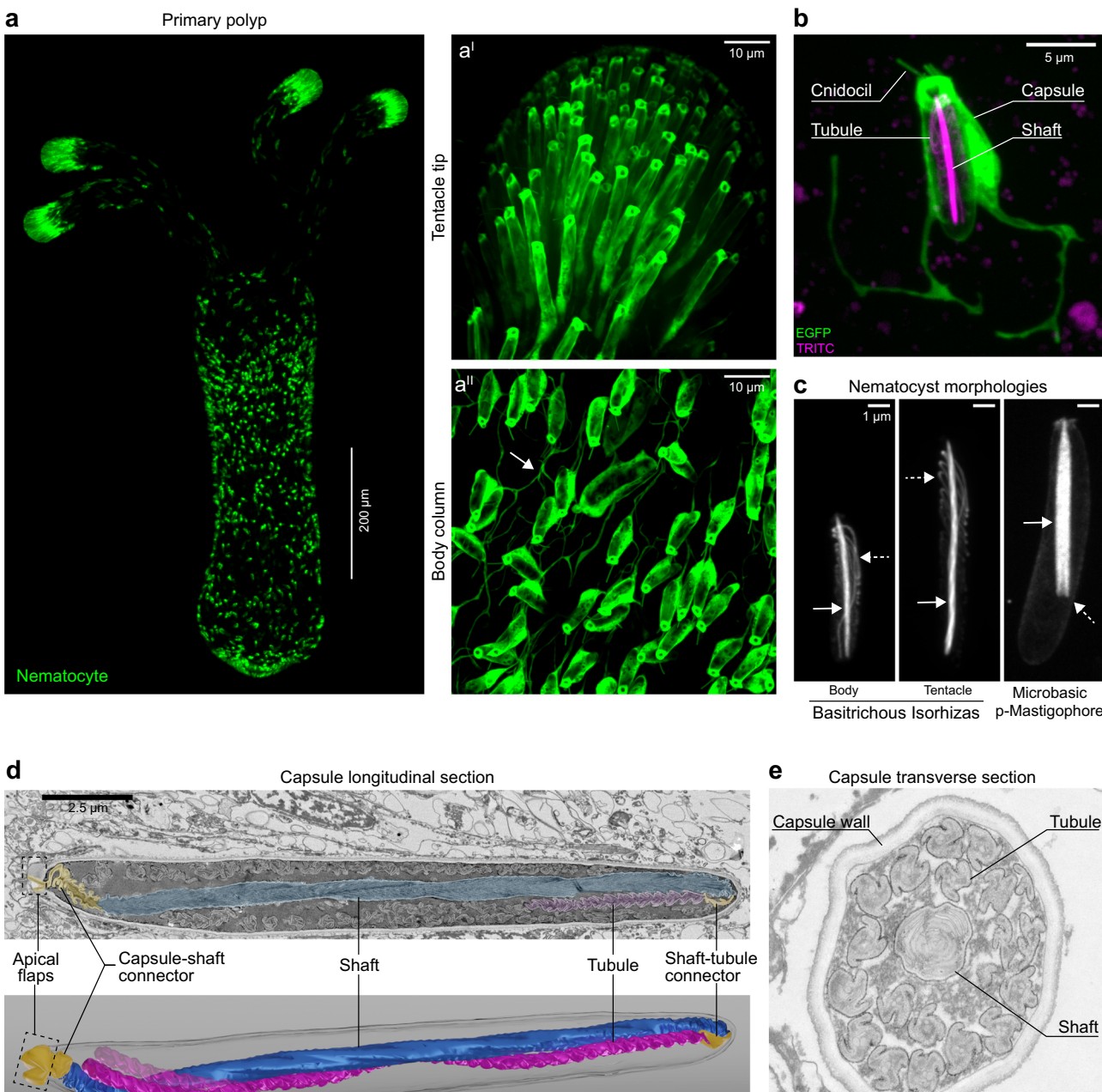

**Fig. 1 The architecture of undischarged nematocysts. a** Nematocytes (*green*) of a transgenic *N. vectensis* primary polyp expressing EGFP under the control of the *nematogalectin* promoter **aI** EGFP expression at the tentacle tip. **aII** EGFP expression on the body column. The arrow points to the network of cellular processes connecting nematocytes. The images are representative of primary polyps tentacles and body columns from 6 spawns. **b** Close-up view of a single nematocyte on the body column of a primary polyp expressing EGFP in nematocytes (green). The cnidocil (sensor) apparatus, cell body, and neurite-like processes are shown. TRITC (magenta) labels the capsule, the centrally positioned shaft, and the folds of the tubule (n = 10 primary polyp body columns, five experiments). **c** Nematocyst morphologies in *N. vectensis* based on the fluorescence signal of incorporated TRITC. Basitrichous isorhizas capsules (n = 19 short, n = 20 long) with densely labeled shaft (arrow) and coiled tubule (dashed arrow) continuous with the compressed shaft (left and middle panels). Microbasic p-mastigophores capsules(n = 10) shaft (arrow) with its distinctive V-shaped notch (right panel, dashed arrow) Images representative of purified nematocysts from ~300 primary polyps. Scale bars 1 µm. **d** Longitudinal section of a nematocyst showing densely coiled shaft filaments (blue), a portion of the coiled tubule (magenta), and the two connector regions, the capsule-shaft connector and shaft-tubule connector (yellow). The apical flaps are seen in a partially open conformation (dashed box). Corresponding 3D reconstruction of the longitudinal section shows the capsule, central shaft (blue), a portion of the attached tubule (magenta), connector regions (yellow), and apical flaps (dashed box). **e** Transverse section of a nematocyst showing the capsule wall, dense lamellar shaft, and the propeller shaped tubule. (n = 2 complete volume capsules images from ~350 capsules visible in 2 primary polyp samples.

found that the compressed shaft consisted of tightly coiled filaments vertically aligned to the capsule aperture formed by the apical flaps (Fig. 1d, box). The filaments were composite structures with stacks of lamellae built from electron-dense and electron-lucent layers. These results confirm a similar triplet lamellar structure to that observed in *Anemonia sulcata* by Godknecht and Tardent (1988), who noted the tip of the shaft is formed by staggered lamellae converging at a small area pointing towards the capsule aperture[17].

Close inspection of our SEM data further revealed that the thread wall encasing the shaft filaments was connected to the apical flaps with a loose capsule-to-shaft connector. A similar shaft-to-tubule connector was located between the basal end of the shaft and the apical end of the tubule. The tubule was twisted, forming pleats in regular lengthwise segments (Supplemental Movies 2, 3). In capsule cross-sections, the shaft lamellae were observed to be tightly coiled and compressed, while the tubule cross-section exhibited a propeller-shaped structure (Fig.1e; Supplementary Movie 4).

**Nematocyst discharge and eversion of the stinging thread**. To determine the distinct phases of thread operation, we next recorded fluorescent high-speed movies of discharge events in TRITC-labeled animals (Supplementary Movies 5–7). Following in situ nematocyte stimulation, the compressed shaft was first ejected as a dense projectile which then rapidly expanded to form an elongated cylinder through which the tubule emerged (Fig. 2a$^{II}$–a$^{IV}$, arrows; Supplementary Movie 5). Based on these observations, we defined three principal phases of nematocyst operation: shaft discharge (Phase I), shaft eversion (Phase II), and tubule eversion (Phase III; Fig. 2, box). Using SEM, we visualized the ultrastructure of the discharging shaft. In the undischarged state, we observed sparse lamellae decorating the region where the shaft tapered to the capsule-shaft connector (Fig. 2b, dashed arrow). During the early stages of discharge, the everted capsule-shaft connector formed a skirt around the traversing shaft, creating a double-walled structure (Fig. 2c, arrow) with the uneverted shaft moving forward inside the connector (Fig. 2c, blue). The everted capsule-shaft connector was externally covered with sparse filaments resembling irregular spines originating from the eversion of the lamellae observed in the undischarged state (Fig. 2c, dashed arrow). Serial SEM sections also captured an everted connector (Fig. 2d, arrow) in which the tubule could be observed departing the capsule (Fig. 2d, dashed arrow; Supplementary Movie 8). Finally, in SEM sections of a partially discharged nematocyst thread, we observed the uneverted tubule traversing inside of its everted fraction, which was decorated externally with hollow barbs (Fig. 2e, arrow; Supplementary Movie 9).

To better understand the structural changes described above, we next analyzed super-resolution images of TRITC-labeled threads undergoing eversion. This approach allowed us to demonstrate the existence of a triple helical geometry of the uncoiled shaft filaments together with the traversing uneverted tubule which could be traced by the labeling of the barbs (Fig. 2f). Interestingly, the thread wall did not incorporate TRITC and was invisible in fluorescent images but could be seen in corresponding SEM cross-sections as an electron-lucent layer that enclosed the uneverted tubule in its compacted state (Fig. 2f, f$^{II}$). The highly ordered arrangement of barbs within the uneverted tubule indicates that these structures are stacked as a column, which appeared as a single filament in fluorescent images (Fig. 2f, f$^{I}$). Further, SEM cross-sections through the shaft showed that its filaments consisted of lamellae that enclosed the traversing uneverted tubule, as seen in fluorescent images (Fig. 2f, f$^{III}$, arrow).

To visualize the thread wall that was otherwise invisible in optical images, we next focused our attention on other components of the thread. Together with minicollagens, the nematocysts of *Hydra* and *Nematostella* contain glycans and show similarities to the extracellular matrix in composition[44–48]. The nematocyst-specific lectin, Nematogalectin, acts as a scaffold linking the minicollagens to glycans, mainly consisting of a non-sulfated chondroitin sheath[30,46,49]. Considering the presence of lectins in the structure, we hypothesized that the presence of GAGs could be detected with fluorescently labeled sugar-binding lectins. Thus, we stained discharged nematocysts with fluorescent dye-conjugated Wheat Germ Agglutinin (WGA), which is selective for GlcNAc chains, and found that WGA strongly bound to the electron-lucent thread wall (Fig. 2g)[50]. Co-staining with WGA and TRITC showed that the WGA-stained material did not co-localize with TRITC, but rather formed a laminate with the TRITC-labeled structures. Finally, images of threads in an early everted state showed that the TRITC-labeled shaft surrounded the WGA-labeled tubule wall. The connector regions lacking filaments or barbs which were poorly labeled with TRITC were more strongly labeled with fluorescent WGA (Fig. 2g$^I$–g$^{III}$).

Importantly, we observed that TRITC labeling overlapped with the shaft filaments in the transillumination channel, suggesting that the shaft fibers harbor TRITC labeled material (Supplementary Fig. 2a, arrows). In contrast, fluorescent WGA labeling was enriched in the interior, surrounding the wall structure. The WGA layer formed repetitive lamellae with the TRITC labeled regions but did not overlap with TRITC labeled material (Supplementary Fig. 2b, c, TRITC, arrows; WGA, dashed arrows). By combining TRITC and WGA labeling with antibody staining of the minicollagen Ncol4, we determined that TRITC signal was present in the threads of the mature capsules only where the thread is invaginated. TRITC did not label maturing capsules, which can be stained with *Nematostella* minicollagen Ncol4 as reported by Zenkert et al. (2010)[24,25]. In tentacle tips, developing capsules deep inside the ectoderm stained with Ncol4 antibody while fully matured capsules lining the surface of the tentacle epithelium were not stained (Supplementary Figs. 3, 4, arrows). In conclusion, these results suggest that the thread consists of two layers: a TRITC-labeled layer forming the TRITC-detectable shaft filaments and barbs, and a WGA-labeled layer forming the overall cylindrical thread wall, including the connector regions, shaft wall, and tubule wall.

**The mechanism of shaft eversion**. Structural studies of nematocyst threads penetrating gel substrates indicate that shaft eversion is initiated at the shaft's apex[17]. To determine how the shaft transforms from its compressed state to a loose triple helical structure, we captured the early stages of discharge by treating *Nematostella* primary polyps with a solution that simultaneously triggers discharge and rapidly fixes the samples[24]. The reconstructed sequence of events from still images revealed the complex geometric transformation of the shaft as it exited the capsule in a coiled configuration (Fig. 2h and Supplementary Fig. 5a, arrows). We found that during shaft discharge (Phase I), the ejected shaft continued to move forward as a dense projectile inside the capsule-shaft connector until the connector extended to its maximal length. During shaft eversion (Phase II), the filaments started to uncoil from the apex of the shaft, turning inside out, thereby everting, while the basal end of the shaft moved forward inside the uncoiling filaments (Supplementary Fig. 5b). The everting tip of the shaft exhibited a spearhead like structure (Supplementary Fig. 5c, arrow). The tubule was attached to the basal end of the shaft through the shaft-tubule connector and was thus pulled through the newly formed lumen inside the uncoiling shaft filaments. This movement resulted in

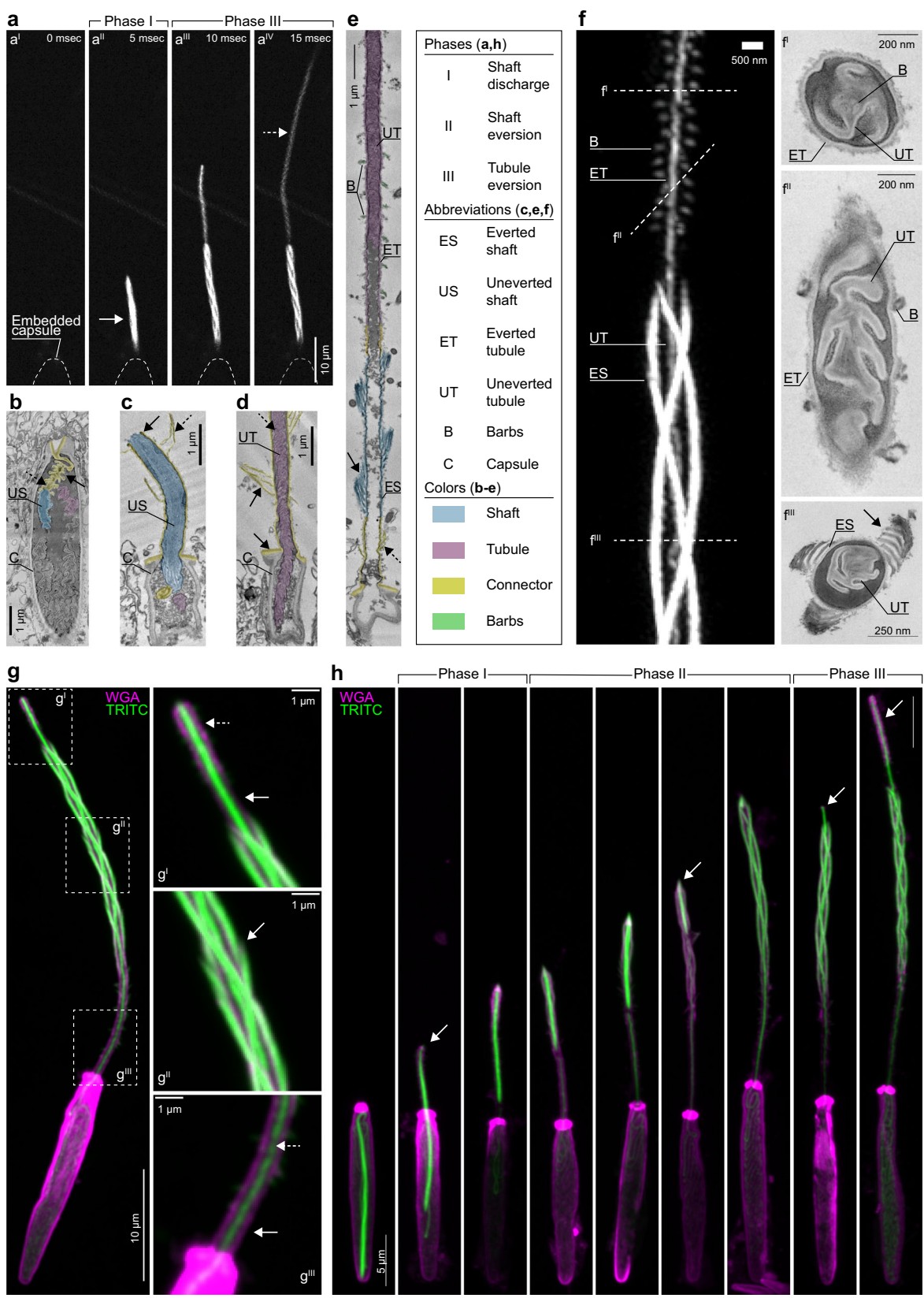

complete eversion of the three filaments where the shaft's former apical end became its basal end. Finally, the everted shaft lumen opened, permitting the movement of the shaft-tubule connector, initiating Phase III. The emergence of barbs on the everted tubule exterior demarcated a boundary between the shaft-tubule connector

and the tubule itself, and thus marked the beginning of tubule eversion (Fig. 2gI; Fig. 2h, last panel. Supplementary Fig. 5a, last panel). Altogether, these data indicate that shaft eversion executes a reproducible series of physical transformations that involve the uncoiling and forward motion of its filaments.

**Fig. 2 Nematocyst discharge and the mechanism of shaft eversion. a** Live imaging of nematocyst discharge. TRITC labeled nematocysts were discharged in vivo and frames were captured at 5 milliseconds intervals. **aI–aIV** Snapshots showing distinct phases corresponding to the shaft discharge and tubule elongation. Note: The shaft eversion (Phase II) was too fast to be captured in this sequence. **b** Longitudinal section of a capsule (C) prior to discharge showing the uneverted shaft (US) and its taper (dashed arrow) to the shaft-tubule connector (yellow, arrow), the apical flaps (V-shaped), and the uneverted tubule (magenta). **c** Longitudinal section of a discharged capsule during shaft eversion. The uneverted shaft filaments (US, blue), capsule-shaft connector (outside the capsule, yellow), a portion of the tubule (magenta), and shaft-tubule connector (yellow).The double-walled structure (arrow) and sparse lamellae on the connector's exterior (dashed arrow) were shown. **d** Longitudinal section of an everted shaft-capsule connector (yellow, arrows) and traversing uneverted tubule (UT, magenta, dashed arrow). **e** Longitudinal section of a partially everted tubule. The everted shaft (ES, blue, arrow), capsule-shaft connector (yellow, dashed arrow), parts of the uneverted tubule (UT, magenta) inside the everted tubule (ET), and barbs (B) are shown. **f** A partially everted nematocyst thread revealed by TRITC incorporation showing the everted shaft (ES) filaments, the uneverted tubule (UT, center) and the barbs (B) decorating the everted portion of the tubule (ET). Corresponding EM cross-sections: **fI** Cross-section of the tubule. Labels indicate the everted and uneverted tubule segments with barbs (B) located at the center. **fII** Oblique section of the partially everted tubule showing tubule segments and barbs at the exterior (B). **fIII** Cross-section of the everted shaft (ES) and the traversing uneverted tubule (UT). The electron-dense and -lucent layers of the ES filaments (*arrow*) were shown. **g** A partially discharged nematocyst stained with fluorescent dye-conjugated wheat germ agglutinin (WGA, magenta) and TRITC (green). Magnified regions: **gI** The shaft-tubule connector (arrow) and the everting tubule emerging with barbs (dashed arrow). **gII** The shaft filaments (green, arrow) and the WGA labeled thread wall (magenta). **gIII** The capsule-shaft connector (arrow) showing the thread wall (WGA, magenta) and the traversing uneverted tubule (TRITC, green, dashed arrow). **h** Representative images of the distinct phases of thread eversion. Fluorescent staining shows the geometric transformations in each phase. Arrows indicate the apex of the thread during distinct phases.

**The mechanism of tubule eversion.** SEM and fluorescent images revealed differences in composition and structure between the triple-helical fibrous shaft and the smooth cylindrical tubule (Fig. 3a, b). While shaft eversion can be explained by the motion of three filaments, the tubule lacks such geometry and likely everts by a mechanism that involves the unfolding and untwisting of the tubule wall[19,29]. Live imaging revealed that during tubule elongation the forward-moving tubule untwisted and relaxed to a cylindrical state (Fig. 2aIII–aIV; Fig. 3b, Stages 1, 2; Supplementary Movie 5). At the everting tip, the helical twists of the everting tubule segment could be seen due to the concentration of WGA staining along the barb pockets (Fig. 3c). In contrast, in an image captured shortly after the initiation of tubule eversion, the tubule was seen as a double-walled cylindrical structure with a lumen between the uneverted and everted walls, suggesting that it likely relaxed and untwisted rapidly (Fig. 3d, arrows). These results indicate that tubule eversion likely occurs in stages involving untwisting of its propeller-like shape to a cylindrical conformation, and that the action of the everted segment untwisting, and relaxing feeds forward the remaining uneverted tubule to the distal tip.

*Hydra* spinalin is a glycine- and histidine-rich protein present in the spine structures on the surface of the shaft in *Hydra* nematocysts[51,52]. To test the role of the centrally stacked barbs in tubule eversion in *Nematostella*, we used shRNA[53,54] to knock down *v1g243188*, previously reported to be a nematocyte-specific gene encoding a spinalin-like product[55]. However, further analysis suggests that *v1g243188* encodes a fibroin-like factor quite distant in sequence composition to *Hydra* spinalin[51,52], and direct orthologs of *Hydra* spinalin were not identified in *Nematostella*[49]. Nevertheless, we found that *v1g243188* knockdown resulted in weakly labeled, thinner shaft filaments and in some cases visibly disrupted the structure of the barbs. The loss of TRITC intensity indicated that a fraction of the dye was also incorporated into the shaft structure, either directly or indirectly due to the presence of *v1g243188*. While the knockdown disrupted the structure of the barbs and their arrangement (Fig. 3e, arrows), this did not appear to affect thread operation. However, loss of *v1g243188* resulted in increased bending of the tubule compared to controls (Fig. 3e, dashed arrow). This observation suggests that *v1g243188* is a component of the thread that plays a role in its structural integrity but not its operation. Further, we noted that the stereotypical helical arrangement of the barbs and their stacked configurations was disrupted in samples exhibiting the strongest phenotypes (Fig. 3f, arrow,

Supplementary Fig. 6a, boxes). Measurement of the TRITC intensity of the shaft structures in discharged threads indicated that TRITC incorporation was reduced following *v1g243188* knockdown, with an approximate 65% reduction in mRNA levels (Supplementary Fig. 6d, e). In discharged nematocysts, the fully everted thread appeared to be an isodiametric tube composed of a WGA-labeled wall equipped with TRITC-labeled barbs and shaft filaments, excluding the connector regions (Fig. 3g, dashed boxes). The barbs decreased in density from the proximal to the distal end of the tubule, and sparsely decorated the distal region (Fig. 3gII–gIV). We hypothesize that the barbs, internally stacked before eversion and externally helically distributed after eversion, might function as a skeleton that prevents further bending and kinking for the elongating tubule. Indeed, as the barbs lessened distally, the tubule appeared to become more prone to kinking compared to the proximal regions which could bend in smooth curves (Fig. 3g, arrow). Interestingly, in live imaging of an elongating thread, we observed that the tubule performed smooth 180° turns in its barb dense proximal region (Supplementary Movie 10). Altogether, our data indicate that tubule eversion involves the unfolding of the tubule wall in which barbs likely provide structural support for the elongating thread.

**Model of the geometric transformation of the shaft and tubule.** These findings allowed us to build a model describing the key aspects of the observed geometric eversion in three phases. Our results suggest that the shaft filaments are attached apically to the capsule flaps, and basally to the tubule via connector regions (Fig. 4a). Phase I: Upon discharge, the shaft is ejected along with the capsule-shaft connector covering the ejecting shaft. A double-walled structure is formed (Fig. 4b, c). Based on still images and movies, shaft eversion occurs after complete ejection of the shaft. Thus, we postulate that the connector accumulates maximal elastic stress when the ejected shaft reaches its maximal distance from the capsule (Fig. 4c). Phase II: Elastic stress on the capsule-shaft connector creates outward forces applied to the apex of the compressed shaft filaments resulting in detachment of the filaments and initiation of the eversion process due to the release of elastic stress within the shaft (Fig. 4d–g, initiation). Upon completion of the sequence, a lumen is formed within the shaft that is protected by the thick filaments (Fig. 4h, end of Phase II). Phase III: The final phase commences with the release of the shaft-tubule connector which everts by folding on itself forming a double-walled structure (Fig. 4i). The uneverted segment of the

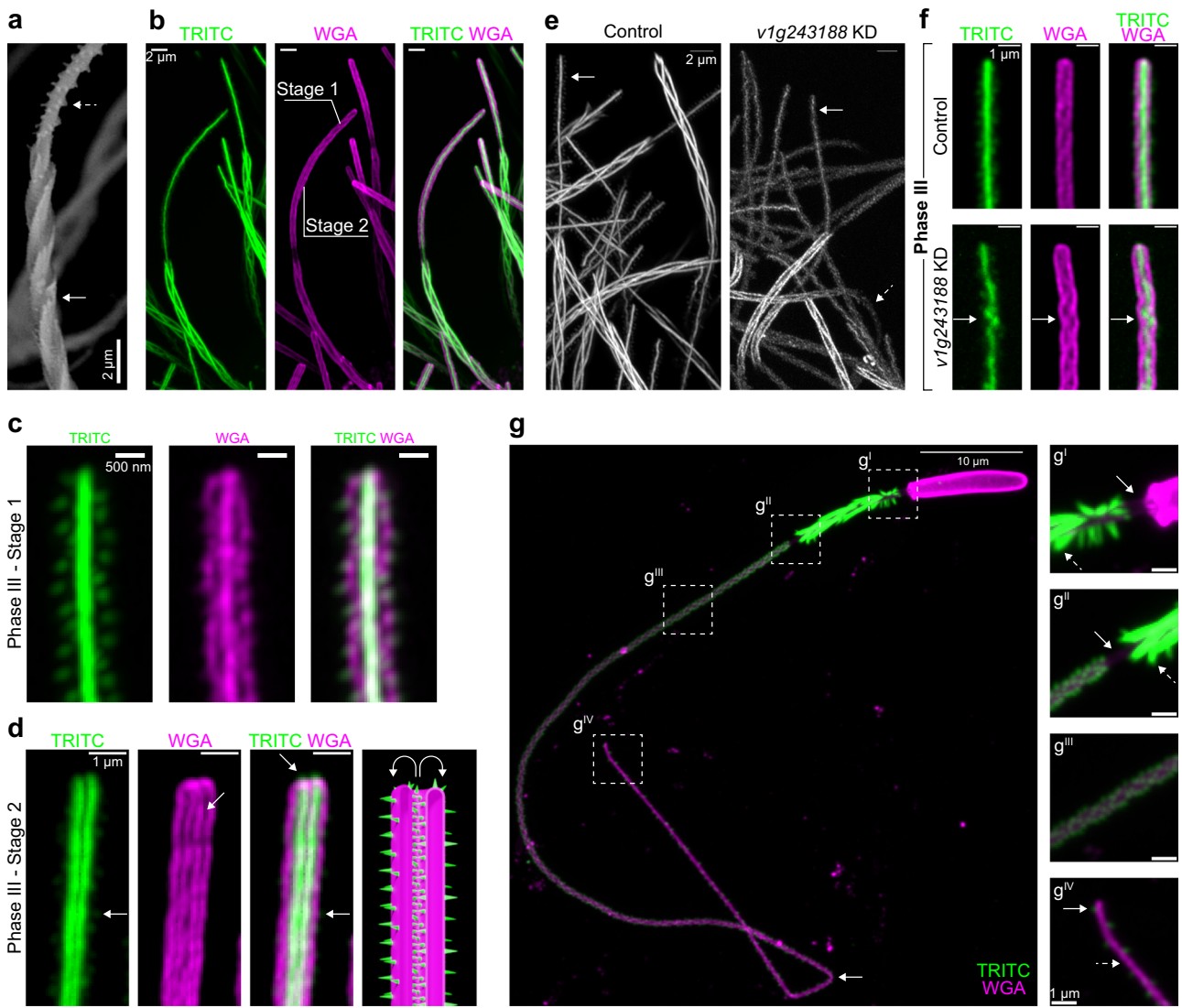

**Fig. 3 The mechanism of tubule eversion. a** SEM image ($n = 6$, 2 experiments) of the shaft (arrow), tubule, and helically arranged barbs (dashed arrow).
**b** Fluorescent image of a partially everted tubule. The shaft filaments and barbs (TRITC, green) and the thread wall (WGA, magenta) are shown ($n = 10$ primary polyp tentacles, 3 experiments). Scale bars 2 µm. **c** Super-resolution image of the everting tubule tip (Stage 1) ($n = 20$ threads of a primary polyp tentacle). The barbs (TRITC, green), the tubule (WGA, magenta), combined channels are shown. Scale bars 0.5 µm. **d** The tip of an everting tubule (Stage 2) ($n = 8/30$ double-walled threads of a primary polyp tentacle). The tubule wall (magenta), its double-walled structure (left middle panel, arrow), and the barbs (right middle panel, arrow) are shown. Illustration of Stage 2 (right panel). Scale bars 1 µm. **e** Structure of the tubule barbs (arrows) in scramble control ($n = 0/20$ partially discharged threads of a primary polyp tentacle) and *v1g243188* ($n = 20/20$ partially discharged threads of a primary polyp tentacle) shRNA KD samples (dashed arrow). Brightness was adjusted to visualize the *v1g243188* threads. Scale bars 2 µm. **f** Effects of the scramble and *v1g243188* knockdown on the barbs (TRITC, green) and tubule wall (WGA, magenta). The arrows indicate the disorganized barbs ($n = 25/25$ threads compared to scramble ($n = 0/25$ threads). Brightness was adjusted to visualize the v1g243188 threads. Representative of partially discharged threads from primary polyp tentacles (Scramble, $n = 27$ primary polyp tentacles; *v1g243188*, $n = 25$ primary polyp tentacles, $n = 5$ shRNA knockdown experiments). Scale bars 1 µm. **g** A fully everted thread labeled with TRITC and WGA ($n = 15$ purified and discharged threads). Arrow indicates kinking at a distal tubule site. **g$^I$** Capsule-shaft connector (arrow) and the everted shaft (green, dashed arrow). **g$^{II}$** The shaft-tubule connector (arrow), and the everted shaft (dashed arrow). **g$^{III}$** Fully everted tubule wall (magenta) and the barbs (green). **g$^{IV}$** Tip of the thread showing the tubule wall (magenta, dashed arrow) sparsely decorated with barbs (green, arrow).

tubule then progressively exits the capsule and moves through the everted portion of the elongating thread (Fig. 4j).

## Discussion

In this article, we described the 3D organization of the nematocyst and the sequence of geometric transformations that occur upon its activation. We also suggest a model explaining the specific mechanisms of thread eversion. Based on our results, we conclude that nematocyst operation occurs in three stages involving a complex transformation of the shaft and the elongation of the tubule, during which energy stored in the overall structure is transformed to kinetic energy. The shaft performs two critical functions: first as a compressed syringe to penetrate the target cuticle; second as a protective tunnel for passage of the thin tubule.

The structure of the shaft of the anthozoan nematocysts exhibits a staggered lamellar structure that differs from the specialized *Hydra* stenoteles which exhibit an arrowhead-like collocated stylet[3,17,56,57]. Godknecht and Tardent have previously

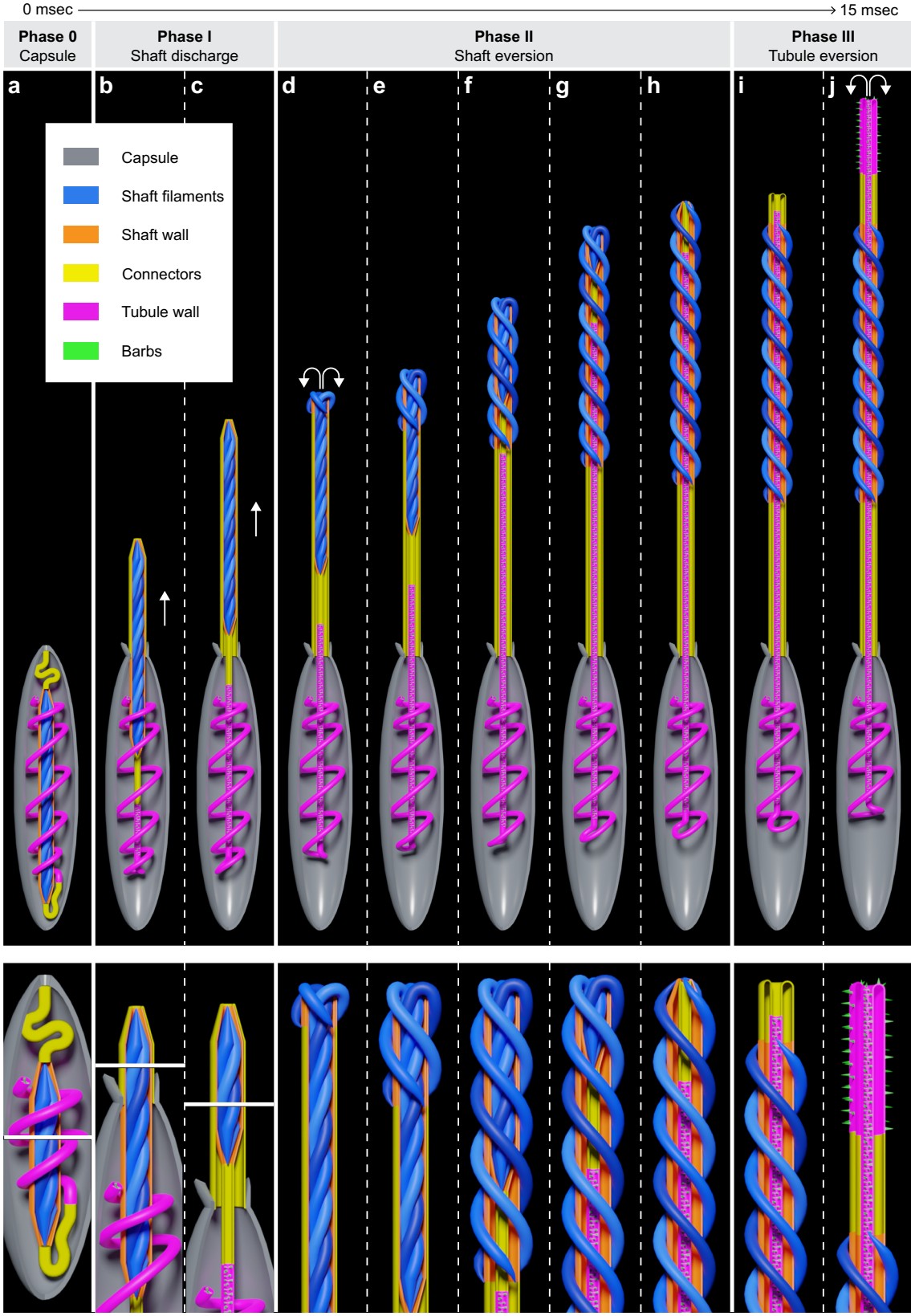

suggested that the staggered arrangement of the lamellae as seen in the shafts of *Anemonia Sulcata* nematocysts results in a "hammer-drill-like" impact on a small area of the target during eversion[17]. In *Hydra* stenoteles, the tip of the stylet impacts and pierces the target at a single point[2,3,11]. Thus, the kinetics of discharge in the *Hydra* stenoteles is orders of magnitude faster than the slower process of shaft eversion we observed in *Nematostella* nematocysts[11,12].

The shaft eversion process resembles the mechanics of a Y-shaped slingshot wherein elastic energy is stored in two bands attached to a pad containing a projectile. Upon release of the pad, the stretched bands experience a geometric eversion. The elastic

**Fig. 4 Model of the geometric transformation of the shaft and tubule.** The box indicates the sub-structures of the nematocyst. Lower panels: Magnified views of critical regions during distinct phases. **a** Undischarged capsule with tightly coiled shaft surrounded by the shaft wall, two connectors, and the coiled tubule. **b, c** Initial stage of shaft discharge (Phase I). The forward movement and eversion of the capsule-shaft (CS) connector enclosing the ejected shaft filaments are shown. The arrows indicate the forward movement of the shaft. **d–h** The geometric eversion mechanism of the shaft and uncoiling of the compressed shaft filaments (Phase II). Arrows indicate the direction of the forces applied to the apex of the compressed shaft filaments. **e–g** Steps in the progression of shaft eversion. The model shows the uncoiling shaft filaments and the forward movement of the shaft-tubule connector and the tubule. **h** The final stage of shaft eversion. Note: The basal end of the uneverted shaft becomes the apical end of the everted shaft. **i–j** The eversion mechanism of the shaft-tubule connector and the tubule (Phase 3).

energy stored in the bands converts into kinetic energy of the accelerating projectile. Nematocysts utilize a similar approach, but due to the limited space inside the capsule store the elastic energy required for shaft eversion by bending and twisting three filaments and attaching them to the capsule and the tubule. The helices of the everted filaments exhibit a larger radius and step compared to their uneverted configuration. Thus, the amount of elastic energy stored in the helical coils decreases in the everted state. The excess energy released during shaft eversion is possibly used for the release of the shaft-tubule connector (Supplementary Note 1, Supplementary Fig. 7). The connectors might function in the initiation of eversion; stretching of the capsule-shaft connector utilizes energy from the discharge to initiate shaft eversion, while the shaft-tubule connector transfers the elastic energy of the "slingshot" to initiate tubule eversion. The loose shaft-tubule connector rapidly forms a cylindrical tube which might act as a buffer zone for the transition of the three-filament shaft to the twisted tubule.

The source of the driving force for tubule penetration could be explained by the osmotic pressure and the elastic forces accumulated in the tubule structure. It has been shown that hydration of ruptured capsules results in extrusion and untwisting of the tubule without undergoing eversion, suggesting that the twisted tubule stores elastic energy to be later transferred into kinetic energy by acting as a spring that is released by relaxation to a cylindrical state[19,20]. The double-walled structure seen upon relaxation (Fig. 3d, second panel, *arrow*) likely allows the flow of PG matrix from the capsule into the lumen, recharging the forces that push the tubule forward[20]. The process is repeated until the tubule is fully elongated or reaches an obstacle (Supplementary Movies 10, 11). In summary, this study demonstrates the operational capability of the nematocyst as a complex and self-assembling biological micromachine. We propose that these ancient and sophisticated organelles represent an ideal model for biologically inspired microscale devices that could be utilized in diverse applications ranging from medical technology to materials science.

## Methods

**Animal husbandry.** Animals were raised at 23 °C in 12 parts per thousand (ppt) artificial seawater (ASW; Sea Salt; Instant Ocean). Spawning induction and de-jellying were carried out as previously described[58]. Embryos and polyps were cultured at either room temperature 23 °C or 25 °C.

**Generation of the *nematogalectin > EGFP* transgenic reporter line.** The transgenic reporter line was generated by meganuclease-mediated insertion of a plasmid containing EGFP under the control of a nematocyte-specific *N. vectensis* nematogalectin promoter[59,60]. This construct was generated as part of a dual reporter system that harbors an mScarlet-I neuronal reporter that was not analyzed here and will be described in a forthcoming publication.

**In vivo labeling of *Nematostella* with (5,6)-tetramethylrhodamine isothiocyanate (TRITC).** Live *Nematostella vectensis* planulae (2 dpf) were allowed to react with the amine reactive rhodamine derivative 5/6-tetramethyl-rhodamine-6-isothiocyanate, TRITC (Cayman Chemical, No. 19593) for a short duration (30 min–1 h). The animals were incubated for 1 h at a final concentration of 1 μM for live imaging of discharge. For fixed specimens, TRITC at a final of 25 μM is

incubated for 1 h with 2dpf larvae. The fluorescent dye stained the animals without any apparent toxicity up to 25 μM concentration tested in this study. Upon incubation, the reactive dye was removed by multiple washes. Animals were transferred to clean dishes in dye-free medium for 3–5 days until mature nematocytes emerged and the non-specific background fluorescence disappeared substantially. Stock solutions of 25 mM TRITC was prepared and stored frozen at −20 °C and used without an observable reduction in the chemical reactivity.

**Electron microscopy.** For scanning electron microscopy (SEM) of topography (Fig. 3a), samples were processed according to previous reports[61]. Briefly, samples were fixed in 2.5% glutaraldehyde and 2% paraformaldehyde in 0.1 M NaCacodylate buffer and stained with aqueous tannic acid, osmium tetroxide, thiocarbodrazide, and then osmium tetroxide again (TOTO). Samples were dehydrated in a graded series of ethanol and critical point dried in a Tousimis Samdri 795 critical point dryer, mounted on stubs, and imaged in a Hitachi TM4000 tabletop SEM at 15 kV with BSE detector. For electron microscopy (EM) of internal ultrastructure, samples were fixed as above, with secondary fixation in 1% buffered osmium tetroxide for 1 h and *en bloc* staining in 0.5% aqueous uranyl acetate carried out overnight at 4 °C. A graded series of ethanol was used for dehydration with acetone as a transition solvent and infiltration in Hard Plus resin (Electron Microscopy Sciences). Samples were cured for 48 h at 60 °C and serial sections were cut at 50 nm using a Diatome 45-degree ultra-diamond knife or an AT-4 35-degree diamond knife on a Leica UC7 ultramicrotome. Sections were collected on slot grids for STEM imaging and a flat substrate (coverslip or silicon chip) for SEM imaging, and post stained using 4% uranyl acetate in 70% methanol for 4 min and Sato's triple lead stain for 5 minutes. Sections on flat substrate were mounted on stubs, the underside of the coverslip painted with silver paint to mitigate charging, and all were coated with 4 nm carbon in a Leica ACE600 coater. Sections were imaged in a Zeiss Merlin SEM using the aSTEM or 4QBSD detector, SmartSEM (6.0.0, Zeiss), and Atlas 5.2.2.15 software (Fibics, Inc.). Serial images were aligned and traced for 3D modeling in IMOD (4.9.10)[62], and 3D models rendered in Blender 2.92 (Blender Foundation). Straightening of fully discharged nematocyte (Fig. 2e) was done in Fiji (ImageJ)[63]. Whole nematocyst capsule image composite (Fig. 1d) and false coloring were done in Photoshop 2021 (Adobe, Inc.).

**Live imaging of nematocyst discharge.** The nematocysts of the TRITC treated animals were discharged by decreasing the pH of the medium using acetic acid. The nematocysts discharge in vivo when the medium becomes acidic. Primary polyps were immobilized in glass bottom dishes by sandwiching between a glass slide and the bottom of a glass bottom dish using silicone sealant. The images captured after dropwise addition of glacial acetic acid (37%) to the ASW which triggers capsule discharge when the pH sufficiently decreases in the medium below a certain threshold. Live imaging of the nematocyst maturation and discharge events were recorded with Yokogawa CSU-w1 spinning disc system on a Nikon Ti2 platform with 100× objective.

**Immunofluorescence.** TRITC treated primary polyps were fixed in Lavdovski's fixative (ethanol : formaldehyde : acetic acid : dH$_2$O; 50:10:4:36) overnight[24]. The fixative was removed, and the samples were washed 5 times in 1 ml PBS pH 7.4 to remove the fixative. Samples were permeabilized with 0.1 % Triton-X100 in PBS, pH 7.4 for 15 min. After several additional washes in PBST (0.1% Tween 20 in PBS, pH 7.4), the polyps were first blocked for 1 h and incubated over night at 4 °C with NvNCol-4 (1:500) in PBST supplemented with 10% goat serum. Minicollagen and Ncol4[24] Anti-NvNCol-4 antibody raised against *Nematostella vectensis* minicollagen NvNcol-4 protein (Rabbit, dilution 1:500) was a kind gift from Suat Ozbek, Heidelberg University). The polyps were washed three times in PBST supplemented with 10% goat serum and incubated with Alexa Fluor 647 coupled anti-rabbit secondary antibody (Goat anti-rabbit IgG (H + L), dilution 1:500, Thermo Fisher, Cat. #: A21245, Lot: 1981173) and WGA-OregonGreen (dilution 1:500, Invitrogen Cat. #: W7024B, Lot: 2298084) overnight in PBST supplemented with 10% goat serum. Thereafter, the polyps were washed several times in PBS and incubated in 90% PBS/Glycerol overnight. The polyps were transferred to a glass slide and mounted on a glass slide with ProLong Glass antifade mounting medium (ThermoFisher, Cat. #: P36982). Fluorescence images were acquired using Yokogawa CSU-w1 on a Nikon Ti2 platform. Super-resolution fluorescence confocal images were acquired using the Zeiss LSM780 in Airyscan mode.

**Super-resolution fluorescence confocal imaging**. In Fig. 2f, TRITC treated primary polyps were fixed in Lavdovski's fixative (ethanol : formaldehyde : acetic acid : dH₂O; 50 : 10 : 4 : 36) overnight[24,25]. After several washes, the polyps were transferred to PBS/Glycerol (PBS) and incubated overnight. The polyps were transferred to a glass slide and mounted with ProLong Glass Antifade Mountant (ThermoFisher, Cat. #: P36982). The polyps were spread onto the glass slides. The labeled tentacles with partially discharged nematocysts were imaged either intact or crushed with the cover slide to detach the capsules from the tissue. The fluorescence imaging was performed using the Zeiss LSM780 in Airyscan mode.

**Purification, discharge, and staining of TRITC labeled nematocysts**. TRITC treated primary polyps were frozen in liquid nitrogen, thawed, and macerated manually with a plastic pestle. The samples were suspended in 1 ml Percoll (50%, v/v; Sigma Cat. #: P1644) in 300 mM sucrose supplemented with 0.01% Tween20 to prevent adhesion to the microcentrifuge tubes. The tissue is further disrupted by pipetting up and down. The mixture is allowed to settle on ice for 30 min and centrifuged for 15 min at 950 g. The pellet is washed twice with PBS with 0.01% Tween-20) resuspended in nematocyst discharge buffer (10 mM Tris, pH 7.5, 10 mM CaCl2). The discharge was initiated by the addition of 1 mM DTT and incubated for 30 min. Upon incubation for 30 min, 1 μg/ml Wheat germ agglutinin conjugated with OregonGreen (dilution 1:500, Invitrogen, Cat. # W7024B, Lot: 2298084) was added to the tube and incubated for 1 h. The stained samples were washed twice with PBST and centrifuged at 1000 × g for 5 min. A loose pellet is seen and suspended in PBS 5 μl aliquots were spread onto glass slides and mounted with ProLong Glass Antifade Mountant. The images were acquired using Yokogawa CSU-W1 spinning disc system on a Nikon Ti2 platform with 100× objective.

**shRNA knockdown**. The short hairpin RNA targeting the putative *Nematostella* genes *v1g243188* and *Nemve1_232014* were synthesized by T7 polymerase reaction, purified using Direct-zol RNA miniprep Plus kit (Zymo Research, Cat. #: R2072). Purified shRNA was microinjected into unfertilized eggs at a concentration of 1 μg/μl or electroporated at a concentration of ~600 ng/μl to 1 μg/μl according to the methods described previously[53,54]. The eggs were fertilized with sperm from wild-type or transgenic males. Following fertilization, the embryos were incubated for 2 days and treated with TRITC. The following primers(Integrated DNA Technologies) were annealed and used as duplex templates for short hairpin RNA synthesis:
*v1g243188* Forward:
TAATACGACTCACTATAGCGGTGGACTCTACTTATTTTCAAGAGAAATA-AGTAGAGTCCACCGCTT and
*v1g243188* Reverse:
AAGCGGTGGACTCTACTTATTTCTCTTGAAAATAAGTAGAGTCCACCGC-TATAGTGAGTCGTATTA
*Nemve1_232014* Forward:
TAATACGACTCACTATAGGCATCGTTACCAGTACAATTCAAGAGATTG-TACTGGTAACGATGCCTT
*Nemve1_232014* Reverse:
AAGGCATCGTTACCAGTACAATCTCTTGAATTGTACTGGTAACGATGCC-TATAGTGAGTCGTATTA
Scramble Forward:
TAATACGACTCACTATAGCAACACGCAGAGTCGTAATTCAAGAGATTAC-GACTCTGCGTGTTGCTT
Scramble Reverse:
AAGCAACACGCAGAGTCGTAATCTCTTGAATTACGACTCTGCGTGTTGC-TATAGTGAGTCGTATTA.

**RNA extraction and qRT-PCR analysis**. Electroporated animals were dissolved in TRIzol Reagent (Ambion, Ref: 15596018; Lot # 254707 and RNA was extracted using the Direct-zol RNA Mini Prep Plus kit (Zymo Research, Cat. #: R2072) according to the manufacturers recommendations. After RNA extraction, cDNA was synthesized with the High-Capacity cDNA Reverse Transcription Kit (Applied Biosystems, Cat. #:4382406, Lot # 0124432). Finally, qRT-PCR was carried out using Luna Universal qPCR master mix (NEB, Cat. #: M3003L, Lot #: 10133023) using the following primers (Integrated DNA Technologies):
*GAPDH*
Forward 5′-GGACCAAGTGCCAAGAACTG-3′
Reverse 5′-GGAATGCCATACCCGTCAG-3′
*v1g243188*
Forward 5′-CCGCCTTATCCTTCGTTGAT-3′
Reverse 5′-ATGCGGTGGACTCTACTTATTG
*Nemve1_232014*
Forward 5′-TGTGAAGGAACGACGATGTG-3′
Reverse 5′-GACCGTTGATGACCTCGATAC-
Quantitative RT-PCR data was analyzed as previously described[64].

**Image analysis and mathematical modeling**. All image analysis was performed with Fiji (ImageJ, 2.1.0/1.53c). The brightness, contrast, and gamma were adjusted manually. The background was subtracted using the subtract background tool in Fiji. In Supplementary Fig. 6b, c, the brightness was increased to better visualize the thread structure. The length measurements for the undischarged and discharged capsules and tubules were performed manually using Fiji (ImageJ) freehand manual tracing tool and the results were discussed in Supplementary information. The mean signal intensity, area, and length of the objects were acquired in Fiji (ImageJ) by using a measurement tool. The results were used in the estimates described in the supplemental information. The estimates were done in Wolfram Mathematica (12.0). In Fig. 4, the model was created in Blender (2.93).

**Statistics and reproducibility**. For the EM micrographs in Figs. 1d, e and 2b–e, serial sections from two different animals were acquired. From 350 capsules observed in the volumes for both animals, we acquired high-resolution images of three complete volumes of undischarged capsules and two partial volumes of undischarged capsules; one complete volume of a discharged capsule in Phase 1, two complete volumes of discharged capsules in Phase 2, and three partial volumes and one complete volume of discharged capsules in Phase 3.

For qRT-PCR verification of target gene knockdown, three independent knockdown experiments were performed with ~200 polyps per sample. Each sample was analyzed in triplicate. Graphs were generated and statistical analyses were performed using GraphPad Prism (9.3.1). Statistical significance was determined using a two-tailed, unpaired students t-test, *p*-values are indicated in the figure legends. Cohen's *d* was used to assess the effect size for all t-test analyses (mean difference between groups divided by pooled standard deviation).

Representative fluorescence images of partially discharged nematocysts were acquired from the tentacles of TRITC-treated animals fixed with Lavdovski's reagent with identical results in 10 independent labeling and discharge experiments. Figure 1a is a representative image of transgene expression observed in animals from six independent spawns. In Fig. 1b, the nematocyte and its capsule was representative of body column nematocytes ($n = 10$ primary polyp body columns, 5 experiments). In Fig. 1c, capsules were representative high-magnification fluorescent images of small ($n = 19$) and large ($n = 20$) basitrichs and p-mastigophores ($n = 10$) purified from ~200 primary polyps. The image sequence in Fig. 2a and the Supplementary Videos 5–7, 10, and 11 were recorded from three independent live discharge experiments.

Figure 2f and Supplementary Fig. 2 were representative super resolution images of the shaft and tubule of partially discharged threads from four independent experiments. In Fig. 2g and h, the discharge sequence was reconstructed from the representative images of partially discharged threads ($n = 67$ partially discharged threads from primary polyp tentacles, three experiments). Figure 3a is a representative SEM image of partially discharged threads ($n = 6$). Figure 3b is a representative image of WGA- and TRITC-stained primary polyp tentacles fixed with Lavdovski's reagent ($n = 10$ primary polyp tentacles, three experiments). Figures 3c, d were representatives of Stage 1 threads (Fig. 3c, $n = 20$) and double walled Stage 2 threads (Fig. 3d, $n = 8$) among the visible threads on a primary polyp tentacle. Figure 3e, f are representative fluorescent images acquired from partially discharged primary polyp tentacles of scramble control shRNA ($n = 0/25$ threads from 1/27 tentacles) and *v1g243188* shRNA ($n = 25/25$ threads from 1/25 tentacles) from 5 knockdown experiments. Figure 3g was a representative image of purified and fully discharged threads from ~300 primary polyps ($n = 15$ threads).

**Reporting summary**. Further information on research design is available in the Nature Research Reporting Summary linked to this article.

## Data availability

Source data are provided with this paper. Original data underlying this manuscript can be accessed from the Stowers Original Data Repository (ODR) at: http://www.stowers.org/research/publications/libpb-1684

The correspondence and materials requests should be addressed to MG2@stowers.org Source data are provided with this paper.

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

## Acknowledgements
We are grateful to Alejandro Sánchez Alvarado, Jay Unruh, Kausik Si, Whitney Leach, Eric Hill, and Subramanian Ramanathan for helpful comments on the manuscript. We are indebted to Molly Simmons for the illustrations of the model. We would like to thank Xia Zhao and Morgan Harwood (Electron Microscopy Core) for the assistance in experiments and the Stowers Reptiles and Aquatics Facility for the animal maintenance. We are thankful to Suat Ozbek, Heidelberg University, for the minicollagen Ncol4 antibody and Ruohan Zhong for help in generating the transgenic animals. The work was funded by Stowers Institute for Medical Research.

## Author contributions
A.K. and M.C.G. concepted this study. A.K., B.R., and M.C.G. wrote the manuscript. A.K., S.M.C., M.M. K.Z.S., and B.R. performed the experiments and analyzed the data.

## Competing interests
The authors declare no competing interests.
