## [Peer Review File · Nature Communications]

The architecture and operating mechanism of a cnidarian stinging organelleREVIEWER COMMENTS

Reviewer #1 (Remarks to the Author):

Diverse methods (e.g., FIB-SEM, mathematical modeling, immunofluorescence, CLSM) were used to provide insights into the ultrastructure and operation of well-known subcellular weapons, namely nematocysts, in a model cnidarian. The manuscript is well written, organized and ready for publication. The figures and supplementary files were gorgeous, straightforward and informative. I found the interpretations/explanations/modeling of the data fully justified, interesting and accurate. I only spotted one tiny typo on line 288 – page 6 (delete “minutes”). I thank the authors for submitting a manuscript so concise and polished that I can safely justify submitting one of the shortest manuscript reviews I have ever written.

This study successfully characterized the underlying details involved in a specific subcellular mechanism in an emerging model organism. Although meticulous and sophisticated in its approach, the results and conclusions from a study like this are unavoidably esoteric and may not appeal to a broad audience. Ejectile organelles/cells, like nematocysts, are also found in verrucobacteria and a wide variety of eukaryotes, so a comparative approach using the methods employed in this study, where possible, are expected to provide even deeper insights into these other systems and one of the most interesting patterns of convergent evolution across the tree of life: the independent origins of ejectile organelles. If I had to provide one criticism, then it would be the absence of any comparative context for this work beyond previous studies of nematocysts in Hydra and a few other cnidarians in the Introduction and Discussion.

Reviewer #2 (Remarks to the Author):

Nematocysts constitute a highly complex miniature weapon consisting of a capsule body and an attached thread, which is expelled in an inside-out-fashion during the extremely fast discharge process. Threads or tubules can be adorned with horn-like spines or barbs (with highest density at their base) that may contribute to the impact of the initial discharge phase to penetrate the skin or cuticle of the prey. In the standard model Hydra, nematocyst structure and discharge mechanics have been extensively studied and there is a significant number of studies detailing nematocyst morphology and mode of action at high visual and, in particular, temporal resolution.

In the present manuscript, the authors use the basitrich type of nematocysts in the anthozoan sea anemone *Nematostella vectensis* to visualize by fluorescence and 3D electron microscopy its packing

and discharge process with a focus on tubule and shaft structures. While I appreciate the excellent imaging of an anthozoan type of nematocyst presented here, I don't see sufficient mechanistic novelty in the study to warrant publication in this journal. In particular, the functional knockdown experiment aimed at analyzing the role of the barbs for discharge mechanics is poorly documented and not convincing. Below I summarized specific points that need to be addressed before a possible revision.

1. The authors created a transgenic line expressing GFP under control of the nematogalectin promoter, which resulted in a cytoplasmic staining of a fraction of nematocytes (mostly basitrichs). To visualize threads, they used a rather unspecific TRITC staining, stating that it has affinity for minicollagens in the thread structure (in fact, the capsule wall is much richer in minicollagens). It is puzzling why they didn't use a nematogalectin-GFP fusion construct, which would have allowed imaging of thread morphogenesis and served as a much more authentic thread marker. Nematocyst tubules and in particular barbs are notoriously "sticky" and can be stained with completely unrelated primary and various secondary antibodies. To define TRITC-labelled structures as a "collagen-rich layer" and WGA-labelled structures as a "GAG enriched layer" with discrete morphological characteristics is therefore misleading.

2. Although the authors define major stages of shaft and thread eversion during discharge, the movies they provide lack temporal resolution to account for a more precise description of the discharge mechanism (see *Toxicon* 54 (2009) 1038–1045). Given the high resolution of the EM images, it is somewhat surprising that the study falls short of the Godknecht & Tardent paper (*Marine Biology*, 1988) regarding the precision of the quantitative and geometric changes accompanying nematocyst discharge. They cite the relevant literature, but refer to the data only in the context of describing capsule and tubule morphology, not the discharge mechanism. In contrast to the data of Godknecht & Tardent, this work lacks a quantitative and physiological analysis of the discharge process as a whole. Godknecht & Tardent clearly showed that there is a doubling of capsule volume (180%) prior to discharge, indicating an osmotic increase in the capsular matrix. It was postulated that this pressure increases the volume of the nematocyst to such an extent that the discharge of the stem continues in a second phase. In this critical second phase, the shaft penetrates the cuticle of the prey, allowing the tube to enter the softer tissue beneath the integument. The staggered arrangement of the lamellae in the evaginating tube causes a hammer-drill-like impact on a very small area on the target surface. This hammer-drill-like motion differs from the discharge of the arrowhead-like collocated stylet in stenoteles of *Hydra*, which have only one optimal point of impact. Therefore, the kinetics of *Hydra* stenotele discharge is orders of magnitude faster (Nuchter et al., 2006; Tardent and Holstein, 1984) than the comparatively slow shaft evagination of anthozoan mastigophores (as described here). The ultra-high speed in *Hydra* stenoteles is generated by the interaction of high pressure and an elastically stretched capsule wall. Unfortunately, the study under review is lacking any comparable kinetic or quantitative morphometric data.

3. The TRITC labelled shaft filaments need to be correlated with phase-contrast and ideally with FESEM images. It is not evident which part of the densely arranged lamellae that decorate the shaft and compose the rotating tip of the evaginating shaft and flip backwards after evagination is stained here.

4. Lines 185-87: "Co-staining with the *Nematostella* minicollagen Ncol4 antibody also revealed an overlap with the TRITC signal, indicating that the barbs are made of minicollagens, and TRITC likely labels, in part, minicollagen rich fibers (Fig. 3e)." In 3e there is no antibody staining shown (3c?). In

Zenkert et al., the Ncol-4-specific antibody was shown to stain the capsule body, not the threads or barbs. This needs to be explained. How does the overall Ncol-4 staining look like?

5. The *Nematostella* putative Spinalin sequence is quite distantly related to the Hydra protein. It is much longer, more cysteine-rich (less histidine-rich) and has a pronounced hydrophobic C-term. Is there any evidence that it is actually part of the barbs or even nematocysts in *Nematostella*? A spinalin antibody staining might be instructive here. What is the evidence that the spinalin mRNA/protein is actually depleted by the shRNA injection? The weaker TRITC staining of the shaft in the spinalin KD condition might be a secondary effect and not indicative of TRITC binding to spinalin. Spinalin can be enriched by an extensive SDS/DTT wash of isolated capsules. This might be instrumental to show depletion of spinalin in the KD condition at protein level. The authors state that spinalin KD disrupts the structure of the barbs as shown in Fig. 3e. The resolution is definitely insufficient to evaluate this. Ideally, FESEM images are needed to demonstrate the effect on the morphology and distribution of the barbs. The sharply bent tubule phenotype in spinalin KDs needs to be quantified. If this observation is a rare event, it cannot serve as an argument that spinalin is “a major component of the thread”. In Hydra, spinalin-rich barbs or spines are attachments on the thread surface, but are not part of the thread structure itself. They are synthesized after the completed thread is invaginated into the capsule lumen. Do the kinks correlate with missing or irregular spaced barbs? Do the authors observe kinked threads in spirocysts (which are free of barbs) of KD animals as well? This would substantiate their argument.

6. The “slingshot” model of the discharge process completely neglects the elastic energy stored in the capsule body itself, which undergoes significant volume changes before and during discharge (see point 2). How do they explain effective tubule release in nematocyst types that lack any shaft structures? Also, in isolated nematocyst samples of *Nematostella*, probably as a consequence of incomplete tubule discharge, shafts are regularly found detached from the capsule body in a still compact, unfolded state (see suppl. figure). Wouldn't the proposed elastic energy stored in the compressed shaft lead to a rapid unfolding of these isolate shafts?

Reviewer #3 (Remarks to the Author):

This is an exemplary study of nematocyst structure and operation with state-of-the-art live-cell imaging, confocal and electron microscopy, and stunning image analysis. The accompanying videos are superb, allowing the viewer to grasp the organization of this marvel of evolutionary engineering. The clips showing the image reconstruction of the nematocyst structure from serial sections and the videos of shaft/tubule extension are remarkable. Wow, yes, I am very pleased to see this as a reviewer. Technically, the paper seems very strong to me, but the accompanying text begins with a significant error and I have a couple of additional points for consideration.

1. Abstract: The nematocyst is described both as an organelle and a cell. It is a cell that contains a harpoon made from modified secretory organelles. The discharge mechanism is a form of explosive exocytosis.
2. Introduction, line 33. This is an unforgivable error. Nematocyst discharge is the fastest biological movement: discharge of the Hydra nematocyst can be completed in less than a microsecond. Nüchter et al. estimated 700 nanoseconds. Tubule eversion is much slower and can take as long as a millisecond, but this secondary process is not very swift at all compared with the mechanisms of fungal spore discharge, pollen ejection, and so on. What makes the nematocyst so awe-inspiring is that initial nanosecond eruption. This should be said very clearly in the Introduction even though the new experiments concern the slower processes of shaft and tubule extension.
3. Introduction. The description of the shaft and tubule eversion would benefit from a diagram right up front. L.51 refers to shaft eversion; l.53-4 states that tubule eversion differs from shaft eversion "as the tubule everts simply by turning inside-out." Eversion means to turn inside-out and if the shaft does this, readers may wonder why the tubule is different? I understand that the mechanics of Phase II shaft eversion and the single phase of tubule eversion are very different, but this may not be clear as explained on l.51-

The mechanisms are clarified later, on l.153- when the authors distinguish between the Phase I process of shaft discharge without any eversion and the Phase II process of eversion. Eversion of a portion of the shaft then proceeds to eversion of the connecting tubule. Maybe Phase I and Phase II of shaft extension should be introduced in the sentence on l.47-49: ". . . piercing the target (Phase I), and later everts (Phase II) to form a lumen . . ."

Otherwise, this is a fantastic piece of work.

RESPONSE TO REVIEWER COMMENTS

please note “Response to Reviewer Figures” appended to the end of this document

Reviewer #1 (Remarks to the Author):

Diverse methods (e.g., FIB-SEM, mathematical modeling, immunofluorescence, CLSM) were used to provide insights into the ultrastructure and operation of well-known subcellular weapons, namely nematocysts, in a model cnidarian. The manuscript is well written, organized and ready for publication. The figures and supplementary files were gorgeous, straightforward and informative. I found the interpretations/explanations/modeling of the data fully justified, interesting and accurate. I only spotted one tiny typo on line 288 – page 6 (delete “minutes”). I thank the authors for submitting a manuscript so concise and polished that I can safely justify submitting one of the shortest manuscript reviews I have ever written.

This study successfully characterized the underlying details involved in a specific subcellular mechanism in an emerging model organism. Although meticulous and sophisticated in its approach, the results and conclusions from a study like this are unavoidably esoteric and may not appeal to a broad audience. Ejectile organelles/cells, like nematocysts, are also found in verrucobacteria and a wide variety of eukaryotes, so a comparative approach using the methods employed in this study, where possible, are expected to provide even deeper insights into these other systems and one of the most interesting patterns of convergent evolution across the tree of life: the independent origins of ejectile organelles. If I had to provide one criticism, then it would be the absence of any comparative context for this work beyond previous studies of nematocysts in Hydra and a few other cnidarians in the Introduction and Discussion.

We thank Reviewer #1 for their comments. To address their concern, we rewrote the introductory paragraph to indicate that the explosive discharge of nematocysts, the fastest phase of the process, is comparable to fungal spore ejection, pollen discharge and the ejection of ballistic organelles of dinoflagellates in lines 36-38. While we considered a broader comparative framework within the discussion and find this an intriguing idea, we did not feel it fit well within the scope of the present paper.

In addition, we deleted the typo “minutes” in the methods section (now line 351).

Reviewer #2 (Remarks to the Author):

Nematocysts constitute a highly complex miniature weapon consisting of a capsule body and an attached thread, which is expelled in an inside-out-fashion during the extremely fast discharge process. Threads or tubules can be adorned with horn-like spines or barbs (with highest density at their base) that may contribute to the impact of the initial discharge phase to penetrate the skin or cuticle of the prey. In the standard model Hydra, nematocyst structure and discharge mechanics have been extensively studied and there is a significant number of studies detailing nematocyst morphology and mode of action at high visual and, in particular, temporal resolution.

In the present manuscript, the authors use the basitrich type of nematocysts in the anthozoan sea anemone *Nematostella vectensis* to visualize by fluorescence and 3D electron microscopy its packing and discharge process with a focus on tubule and shaft structures. While I appreciate the excellent imaging of an anthozoan type of nematocyst presented here, I don't see sufficient mechanistic novelty in the study to warrant publication in this journal. In particular, the functional knockdown experiment aimed at analyzing the role of the barbs for discharge mechanics is poorly documented and not convincing. Below I summarized specific points that need to be addressed before a possible revision.

Response: *We appreciate Reviewer #2's detailed comments and have endeavored to address all points with new data and changes to the text. In particular, we would like to thank the reviewer for pointing out much of the foundational knowledge of nematocyst biology. We agree that our original manuscript may not have sufficiently emphasized prior knowledge of nematocyst morphology and function. This was in part due to word limitations on the original submission. Accordingly, we have now rewritten several sections to reference previous literature and extended the manuscript to provide further detail on prior molecular, histological, and live imaging studies.*

*Interpretation of our results was aided by previous efforts to reconstruct nematocyst structure and function by numerous researchers. We aimed at adding upon the seminal work done on the biomechanics and molecular composition of the nematocyst by Holstein, Tardent and Ozbek and their colleagues, which is the foundation for our work. Using *Nematostella*, we confirmed and extended the original models of the operating mechanisms of the nematocysts with new techniques available in the system including gene KD, transgenesis, reverse genetics, and fluorescence microscopy. Indeed, one of the key technical aims of this study was to analyze how complex organelles form and function by exploiting reverse genetics analysis accompanied with versatile imaging assays.*

Reviewer 2 Point 1 Comment 1: The authors created a transgenic line expressing GFP under control of the nematogalectin promoter, which resulted in a cytoplasmic staining of a fraction of nematocytes (mostly basitrichs). To visualize threads, they used a rather unspecific TRITC staining, stating that it has affinity for minicollagens in the thread structure (in fact, the capsule wall is much richer in minicollagens).

Response 2 Point 1 Comment 1: *We observed that the Rhodamine based TRITC dye accumulates specifically on the thread structures at the late stages of capsule wall maturation and only after the thread is invaginated (see Supplementary Fig. 1a, arrows) but not in maturing nematocysts (Supplementary Fig. 1a, dashed arrows, Supplementary Fig. 3, 4, arrows). These results are now discussed in lines 95-98. We postulate that TRITC is incorporated into cross-linked minicollagen fibers in a conformation that cannot be stained with antibodies. At later stages, it is possible that the dye incorporates into the fibers assembled in a specific geometry. The nature of this material that incorporates the dye would be very informative to understand the composition of the thread. Holstein et al. Nature (1994) reported that the innermost section of the nematocyst wall was composed of a thin layer of minicollagen fibers that polymerize to form the inner capsule sheath converging in a single point which was shown to form a whorl. Indeed, TRITC does weakly label the inner wall of the capsule described as interna by Godknecht and Tardent (1988) and Holstein et al. Nature (1994) (Response to Reviewer Fig. 8c, dashed arrows). The signal emanating from the inner wall is weaker than in the thread but visible when the brightness is adjusted. Based on EM images, the shaft filaments comprise repetitive layers of lamellae formed by the same inner wall material that is continuous with the shaft lamellae, thus the signal is exponentially stronger. In many TRITC labeled capsules, we*

observe one or more intensely labelled bright dots (Response to Reviewer Fig. 8a,c, arrow; Supplementary Fig. 6a, arrow). This spot was visible only in the inner capsule wall and was not co-localized with the WGA-stained material suggesting that the dye binds to a distinct material, and perhaps labels the whorl described by Holstein et al. (1994) marking a bright region (Response to Reviewer Fig. 8a-c, dashed arrows, arrows).

Reviewer 2 Point 1 Comment 2: It is puzzling why they didn't use a nematogalectin-GFP fusion construct, which would have allowed imaging of thread morphogenesis and served as a much more authentic thread marker.

Response 2 Point 1 Comment 2: Our study was designed to understand the geometric changes of the thread during its evagination, and we did not pursue a fusion protein strategy in order to avoid the potential functional impairment of Nematogalectin. Labeling the thread with a GFP fusion protein also presents a limitation on fixation methods due to quenching of GFP fluorescence. Thus, we used fixed samples and the more versatile dye labeling approach while circumventing the technical difficulty of visualizing GFP fusions under our fixation conditions. Notably, we have also incorporated a sentence to refer to the previous findings on thread morphogenesis and the role of nematogalectin in thread assembly, which led us to use the nematogalectin promoter for cell-type specific labeling of nematocytes (lines 77-81).

As a general tool, we also point out that the transgenic line we generated will be of broad utility for the study of nematocyte biology. For example, EGFP does not penetrate the capsule and the tubule in transgenic animals. It is thus possible to monitor capsule and thread morphogenesis by contrast with the surrounding GFP+ cytoplasm (Supplementary Fig. 1a, dashed arrows). To illustrate the utility of using an EGFP transgene for analysis of nematocyte development, we further note that the revised manuscript includes shRNA knock-down of a nematogalectin-like gene *Nemve1_232014* and shows the corresponding reduction of *Nemve1_232014* mRNA (Supplementary Fig. 1b-c). The knockdown of this gene in our transgenic background resulted in GFP+ nematocytes harboring multiple vesicles rather than stereotypical capsules and they were unable to form mature capsules (Supplementary Fig. 1b, dashed boxes, arrow vs dashed arrows). These new data are discussed in the revised manuscript on lines 98-104.

Reviewer 2 Point 1 Comment 3: Nematocyst tubules and in particular barbs are notoriously “sticky” and can be stained with completely unrelated primary and various secondary antibodies. To define TRITC-labelled structures as a “collagen-rich layer” and WGA-labelled structures as a “GAG enriched layer” with discrete morphological characteristics is therefore misleading.

Response 2 Point 1 Comment 3: We agree with this comment and rewrote the section to describe the labeled structures as TRITC-labeled and WGA-labeled (lines 193-195). We also edited the line 166 and changed the word “non-collagenous” to “other” components of the thread. To provide further evidence that the WGA and TRITC staining are specific, we also generated super resolution images showing that these structures do not overlap but rather form distinct layers. In the shaft, TRITC and WGA-488 staining differentially labeled two distinct layers (Supplementary Fig. 2b-c, TRITC, arrows; WGA, dashed arrows), forming laminated structures comparable to the electron-lucent and electron-dense lamellae seen in Fig. 2^{fl}. Further supporting the distinction between these staining methods, TRITC (Supplementary Fig. 2c, arrows) labels the barbs whereas WGA (Supplementary Fig. 2c dashed arrows) labels the base of the barbs and predominantly the thread wall. These new data are discussed in the revised manuscript in lines 181-186.

To demonstrate the existence of separate layers of WGA and TRITC in the capsule, we tried another approach. Adamczyk (J. Bio Chem. (2010)) reported that a non-sulfated chondroitin is present in cnidarian organelles. We therefore treated discharged capsules with chondroitinase ABC from Proteus Vulgaris. This data is now included in (Reviewer Fig 8). Upon treatment, the capsule outer wall was partially digested and became thinner but not the inner wall of the capsule. We do not rule out that the reaction conditions rather than the enzyme treatment could result in thinning of the WGA-stained layer. We found a sample where this treatment detached the WGA labeled outer capsule wall from the inner wall which was labeled with TRITC (Reviewer Fig. 8b, c). In this sample, the differential enrichment can also be seen at the apical flaps where the interior of the apical flaps is exclusively labeled with TRITC that adorn a WGA-stained base (Reviewer Fig. 8a, apical flaps). We also noted that the inner wall, stained with TRITC in most nematocysts harbor one or more bright dots (Reviewer Fig. 8a, arrow). The results showed that the TRITC labels a layer distinct from that of the WGA-stained layer in the nematocyst structure.

In sum, based on our findings, we conclude that WGA-488 and TRITC labeling reveal two predominantly distinct materials that form a composite when the outer and inner wall detached from each other. (Reviewer Fig. 8b, c double arrow, dashed arrow). Regarding concerns over antibody specificity, please refer to Response 2.4.2.

Reviewer 2 Point 2: Although the authors define major stages of shaft and thread eversion during discharge, the movies they provide lack temporal resolution to account for a more precise description of the discharge mechanism (see Toxicon 54 (2009) 1038–1045). Given the high resolution of the EM images, it is somewhat surprising that the study falls short of the Godknecht & Tardent paper (Marine Biology, 1988) regarding the precision of the quantitative and geometric changes accompanying nematocyst discharge. They cite the relevant literature, but refer to the data only in the context of describing capsule and tubule morphology, not the discharge mechanism. In contrast to the data of Godknecht & Tardent, this work lacks a quantitative and physiological analysis of the discharge process as a whole. Godknecht & Tardent clearly showed that there is a doubling of capsule volume (180%) prior to discharge, indicating an osmotic increase in the capsular matrix. It was postulated that this pressure increases the volume of the nematocyst to such an extent that the discharge of the stem continues in a second phase. In this critical second phase, the shaft penetrates the cuticle of the prey, allowing the tube to enter the softer tissue beneath the integument. The staggered arrangement of the lamellae in the evaginating tube causes a hammer-drill-like impact on a very small area on the target surface. This hammer-drill-like motion differs from the discharge of the arrowhead-like collocated stylet in stenoteles of Hydra, which have only one optimal point of impact. Therefore, the kinetics of Hydra stenotele discharge is orders of magnitude faster (Nuchter et al., 2006; Tardent and Holstein, 1984) than the comparatively slow shaft evagination of anthozoan mastigophores (as described here). The ultra-high speed in Hydra stenoteles is generated by the interaction of high pressure and an elastically stretched capsule wall. Unfortunately, the study under review is lacking any comparable kinetic or quantitative morphometric data.

Response 2 Point 2: *To address this point we added a paragraph to the Discussion summarizing the Reviewer's description of the discharge mechanism of Hydra stenoteles as described in Toxicon 54 (2009) 1038–1045 as well as the hammer-drill hypothesis of Godknecht and Tardent (1988) (lines 295-302). We specifically indicated the discrepancy between the speed of discharge in Hydra stenoteles vs anthozoan nematocysts in lines 300-302. In the present study we did not have comparable temporal resolution due to the application of high-*

resolution fluorescent imaging approaches, which were essential to describe geometric transformations of the thread movements. As one approach to overcome this limitation, we utilized the Lavdovski fixation method that we adopted from Zenkert et al. Plos One (2011) to capture the earliest stages of discharge in detail. An example of the discharge of nematocysts in their earliest stages on tentacles can be seen in (Reviewer Fig. 1).

We also did not study the impact of the shaft on a substrate described in Tardent and Godknecht due to the limitations of fluorescent confocal imaging. We confirm that the shaft's tip during eversion resembles that of a compact spike in our fluorescent images (Supplementary Fig.5c, arrow). Since the shaft was discharged without impact, we mainly focused on the shaft's eversion based on our fixed images, thus we cannot rule out the hammer-drill hypothesis with our approach.

We have also now inserted a sentence to indicate the shaft is formed by staggered lamellae as seen by Godknecht and Tardent (lines 119-122). We did not observe the ordered detachment of the tile-like arrangement of the shaft lamellae, which would theoretically be visible in fluorescent images. In discharged purified capsules we observed that the lamellae only detached following prolonged treatment with reagents such as DTT. Cracks in shaft filaments were observed upon discharge, especially in the presence of calcium ions. In this case, the lamellar triplets as described by Godknecht and Tardent already form disordered cracks in overall shaft structure (data not shown).

Finally, for the Reviewer's reference, we have included SEM images of the shaft showing its drill bit-like shape after discharge without any signs of delamination of its fibers (Reviewer Fig. 2). We do not rule out the fact that the shaft impact could lead to delamination and detachment of the fibers. In our SEM samples of discharged nematocysts, we observed only a drill bit-like shaft filaments corresponding to a fibrous structure formed by a triple helical arrangement of fibers as seen in our fluorescent and transillumination images of fixed samples and fluorescent live imaging results (Reviewer Fig. 2, arrows; Supplementary Fig. 2a, left most panel; Supplementary Videos).

For the capsule volume measurements, we were technically unable to acquire images at the speed or intensity required to clearly capture the volume change. However, we agree with the volume measurements from *Anemonia Sulcata* reported in Godknecht and Tardent (1988). We added a paragraph describing these observations previously reported on the role of pressure and capsule elasticity (lines 39-51).

Reviewer 2 Point 3 Comment 1: The TRITC labelled shaft filaments need to be correlated with phase-contrast and ideally with FESEM images.

Response 2 Point 3 Comment 1: To address this concern, we have now provided a super-resolution fluorescent image of the thread's capsule, shaft and parts of the tubule stained with TRITC and WGA-488 dye and the corresponding transillumination channel image (phase-contrast was not available). Analysis clearly shows that the helically arranged shaft filaments overlap with the TRITC labeled material (Supplementary Fig. 2a, arrows). The data is discussed in lines 181-183.

Reviewer 2 Point 3 Comment 2: It is not evident which part of the densely arranged lamellae that decorate the shaft and compose the rotating tip of the evaginating shaft and flip backwards after evagination is stained here.

Response 2 Point 3 Comment 2: To show the rotating tip of the evaginating shaft and its backward flip in detail we now provide a new high-resolution sequence of shaft movements in TRITC labeled threads in (Supplementary Fig. 5a). In this response, we also include an image showing the “fireworks” like effect of the rapid fixation that we utilized to capture these movements (Reviewer Fig. 1). We now include a supplementary figure with the sequence of images for the shaft eversion at its apical end and the backward flip (Supplementary Fig. 5b, Phase II, Shaft eversion) Here, the shaft eversion sequence at its rotating tip can be seen in the now included (Supplementary Fig. 5b, Phase II). To further address the reviewer’s comment, we also now provide a super-resolution image showing the rotating tip of the shaft filaments (Supplementary Fig. 5c, arrow). This data is described in lines 206-210.

Reviewer 2 Point 4 Comment 1: Lines 185-87: “Co-staining with the Nematostella minicollagen Ncol4 antibody also revealed an overlap with the TRITC signal, indicating that the barbs are made of minicollagens, and TRITC likely labels, in part, minicollagen rich fibers (Fig. 3e).” In 3e there is no antibody staining shown (3c?).

Response 2 Point 4 Comment 1: We apologize for the typo in Fig. 3e. We deleted the sentence and corrected Fig. 3c. See Response 2 Point 4 Comment 2:

Reviewer 2 Point 4 Comment 2: In Zenkert et al., the Ncol-4-specific antibody was shown to stain the capsule body, not the threads or barbs. This needs to be explained. How does the overall Ncol-4 staining look like?

Response 2 Point 4 Comment 2: To follow up on Reviewer 2’s concerns about Ncol-4 antibody specificity, we have now performed our antibody staining protocol with an unrelated primary antibody - a rabbit polyclonal antibody targeting the Nematostella VASA protein. We observed non-specific labeling of the barbs with the VASA antibody (Reviewer Fig. 3a, a', dashed box) and thus have now removed the observation that Ncol4 antibody labels the barbs from Fig 3c. We now provide a supplementary image of anti-Ncol4 staining in Nematostella tentacles, which confirms the staining pattern reported by Zenkert et al Plos One (2011). We inserted lines 186-192 to describe the Ncol4 antibody staining results (Supplementary Fig. 3, 4, arrows). Similar to what they observed, Ncol4 was found to be exclusively labeling the maturing capsules but not fully mature capsules harboring the thread, which was brightly labeled with TRITC (Supplementary Fig. 3, 4, arrows).

Reviewer 2 Point 5 Comment 1: The Nematostella putative Spinalin sequence is quite distantly related to the Hydra protein. It is much longer, more cysteine-rich (less histidine-rich) and has a pronounced hydrophobic C-term. Is there any evidence that it is actually part of the barbs or even nematocysts in Nematostella? A spinalin antibody staining might be instructive here.

Response 2 Point 5 Comment 1: We added a short paragraph (lines 237-243) describing this discrepancy and agree that the protein described as “spinalin” or v1g243188 is quite distantly related to the Spinalin present in Hydra. Previous reports indicate that Hydra Spinalin is not present in Nematostella (Shpirer et al. BMC evo. Biol. (2014)). We initially sought to investigate the protein described as “spinalin” or v1g243188, which was reported to be expressed in nematocytes reported by Sebé-Pedrós et al Cell (2018), to demonstrate the utility of our methods and understand the role of proteins involved in nematocyte biology in a small shRNA-

based screen. To adhere with the original publication, we kept the original nomenclature, which caused the misunderstanding. To address the reviewer's concern, we now use the gene name v1g243188 throughout the manuscript instead of "spinalin".

Reviewer 2 Point 5 Comment 2: What is the evidence that the spinalin mRNA/protein is actually depleted by the shRNA injection?

Response 2 Point 5 Comment 2: To address this concern we included a new supplementary figure with images of the shaft and the beginning of the tubule in v1g243188 KD along with corresponding measurements of decreased v1g243188 mRNA levels via qPCR (Supplementary Fig. 6). The new data is discussed in line 255.

Reviewer 2 Point 5 Comment 3: The weaker TRITC staining of the shaft in the spinalin KD condition might be a secondary effect and not indicative of TRITC binding to spinalin.

Response 2 Point 5 Comment 3: We agree with the reviewer that the TRITC dye might not incorporate directly, and the reduced dye incorporation might be due to another effect of the knockdown on the thread. We revised the text to indicate the resulting phenotype clearly as loss of dye binding (lines 253-256). However, we believe this loss of reduction in dye incorporation is a good read-out of the effect of the KD, otherwise it would be difficult to study with conventional light microscopy.

v1g243188 KD did not have an effect on the development of animals and nematogenesis. We observed that nematocytes develop and discharge normally in KD samples (data not shown). We do not know if v1g243188 binds to TRITC or incorporates in the thread structure but the loss of the TRITC signal indicates that the presence of this protein is important in the structure of the thread. We revised the text to indicate this observation (lines 244-246). The v1g243188 KD effect is most pronounced in the shaft filaments which results in the visible reduction of TRITC incorporation. WGA dye binds to the thread structure in a different way and its staining protocol is quite different than TRITC labeling. We measured both TRITC and WGA labeling intensity in the shaft regions of the purified and discharged threads in control and v1g243188 knockdown samples and quantified the TRITC incorporation as well as WGA staining intensity in the shaft filaments in knockdown samples (Supplementary Fig. 6d).

Reviewer 2 Point 5 Comment 4: Spinalin can be enriched by an extensive SDS/DTT wash of isolated capsules. This might be instrumental to show depletion of spinalin in the KD condition at protein level. The resolution is definitely insufficient to evaluate this. Ideally, FESEM images are needed to demonstrate the effect on the morphology and distribution of the barbs.

Response 2 Point 5 Comment 4: As stated in Response 2 Point 5 Comment 1, we agree with the Reviewer on the identity of this protein and changed the name "spinalin" to v1g243188 as described above. Thus, we did not perform the extensive SDS/DTT washes of the purified capsules.

Reviewer 2 Point 5 Comment 5: The sharply bent tubule phenotype in spinalin KDs needs to be quantified. If this observation is a rare event, it cannot serve as an argument that spinalin is "a major component of the thread".

Response 2 Point 5 Comment 5: We provide a representative image of v1g243188 knockdown examples vs scramble control shRNA in discharged tentacles that provides numerous threads in distinct discharge states for the reviewer to evaluate the effects of

knockdown (Reviewer Fig. 4 and 5). The tips of several single threads were magnified for observation (Reviewer Fig. 5, dashed boxes, right panels). Note that the knockdown sample has increased background as the brightness needed to be adjusted due to the weaker signal. We quantified, using autocorrelation, the apparent thread straightness and found that threads in v1g243188 knockdown animals were significantly less straight than the control counterparts (Reviewer Fig.6). We also acquired and provided high-resolution images of the thread's shaft in scramble control and v1g243188 KD samples (Supplementary Fig. 6a-c, insets).

Reviewer 2 Point 5 Comment 6: Do the kinks correlate with missing or irregular spaced barbs? Do the authors observe kinked threads in spirocysts (which are free of barbs) of KD animals as well? This would substantiate their argument.

Response 2 Point 5 Comment 6: *This is an interesting point. The kinks frequently occur at the distal regions of the normal nematocyst threads as shown in Figure 3 (Fig. 3g, arrow). In regions with densely spaced barbs, the thread bends in smooth curves rather than kinks as shown in Supplementary Video 10 and discussed in lines 264-266.*

We were unable to label spirocysts with our staining method and would not be able to conclude that the spirocysts lacking any barb like structures harboring TRITC labeled material. The differences in labeling proficiency can provide very interesting information about material compositions of distinct types of cnidae.

Reviewer 2 Point 6 Comment 1: The “slingshot” model of the discharge process completely neglects the elastic energy stored in the capsule body itself, which undergoes significant volume changes before and during discharge (see point 2).

Response 2 Point 6 Comment 1: *We agree with the reviewer on the importance of the elastic energy sourced from the capsule body itself and the osmotically generated pressure. To address this point, we revised our manuscript to indicate that the source of the energy for the shaft discharge and incorporated a paragraph (lines 39-45) describing the previous results to clarify the energy sources for penetration.*

The slingshot model describes how a triple helical shaft structure stores energy and remains in its coiled state through the function of “connectors”, and later uncoils and releases stored elastic energy to initiate the next phase of the events: the release of the shaft-tubule connector and the tubule. The initiation of the tubule operation phase requires a transition from the eversion mechanics predominated by a triple-filamentous structure to a tubular one through a second “tubule-connector” region that keeps shaft in a tightly coiled state.

The supplementary information describes the shaft as a coiled structure which uncoils and expands during eversion. From EM images and WGA staining we conclude that the thread wall is continuous with the capsule wall, and it could possess elastic properties. This suggests that the shaft could store elastic energy when wound. The shaft's compositional similarity to the capsule wall and its operation suggests that the uncoiling releases elastic energy stored in this structure. As we agree upon the source of the energy originating from the capsule, we focused on the geometric movements in the later phases of the discharge, mainly the shaft eversion which likely requires the energy stored in the shaft and the tubule to operate.

Reviewer 2 Point 6 Comment 2: How do they explain effective tubule release in nematocyst types that lack any shaft structures?

Response 2 Point 6 Comment 2: *In this manuscript, we have studied the operating mechanism of the basitrich type of nematocysts in Nematostella and we currently do not have knowledge of the operation of other nematocyst types, other cnidae including the non-penetrants, or the nematocysts harboring darts. We agree that an evertible tubule driven by a pressure and elastic capsule wall is universal among cnidae. The additional functionality, which we describe in the shaft structure here in this study could be the differing innovation that not only changes the form but also the function of the organelle. For this reason, our study, or the hammer-drill hypothesis of Godknecht and Tardent would not be applied to explain the operation of the nematocysts without the shaft structures. The simple and versatile techniques that we developed in this manuscript can be easily applied to other cnidarian model organisms to understand the operation of other nematocysts.*

Reviewer 2 Point 6 Comment 3: Also, in isolated nematocyst samples of Nematostella, probably as a consequence of incomplete tubule discharge, shafts are regularly found detached from the capsule body in a still compact, unfolded state (see suppl. figure).

Response 2 Point 6 Comment 3: *In the reviewer's data figure, the connector was broken, thus detached from the shaft's apical end before reaching its maximum distance from the capsule (Reviewer Fig. 7a, connector). The shaft was unable to open and initiate eversion from where it was attached to the connector. The data supplied by the reviewer appears to be a p-mastigophore with a dart-like tip and a v-notch, which we did not study in our experiments extensively (Reviewer Fig. 7a). However, the data can be regarded as an example of a malfunctioning capsule-shaft connector assuming p-mastigophores are attached to the connector, proximal to the capsule below the tip of the dart. Thus, the shaft's apical end was attached to the capsule through the connector. When the shaft was ejected, the connector must stretch to its maximal distance to initiate eversion of the shaft from its apical end (below the tip of the dart) which did not occur because the connector was detached from the shaft's apex).*

Reviewer 2 Point 6 Comment 4: Wouldn't the proposed elastic energy stored in the compressed shaft lead to a rapid unfolding of these isolate shafts?

Response 2 Point 6 Comment 4: *To address this question, we attached a figure with an image that we acquired with a malfunctioning shaft that uncoiled without eversion (Reviewer Fig.7b). In this image, the capsule was broken. The shaft is seen to uncoil without eversion inside the shaft wall, where the shaft's wall expands to 1 μ m diameter from its normal 500 nm diameter seen in undischarged states (Reviewer Fig.7b dashed line). The basal ends of two shaft filaments became visible and the connector regions expanded conically (Reviewer Fig.7b, Shaft filaments, connectors). Overall, the relaxation of the shaft filaments without eversion resulted in the expansion of the diameter of the shaft wall as well as the connectors.*

This image shows that the shaft wall has elastic properties as it expands and increases the diameter of the shaft wall by two-fold. The stretching of the connectors show that the connectors are also expandable. We propose that the connector regions that we describe in our manuscript as well as the shaft wall are critical to prevent the shaft from uncontrolled operation and spontaneous uncoiling without eversion. The question is how does the shaft keep in its tightly coiled state before discharge? To withstand uncoiling of the shaft, the thread wall must be extremely stiff. Connectors, which we propose as novel, essential features of the structure, may play a critical role as lack of the connector or a broken connector would result in malfunctioning of the stereotypical firing sequence.

We acquired this image of a burst capsule from the samples that we treated with chondroitinase which likely weakened the shaft wall to a point that the tightly coiled shaft filaments uncoiled without the wall's resistance. This suggests that the shaft indeed stores elastic energy. The shaft wall and the connectors are likely made of a very stiff material that keeps the filaments in their dormant coiled state. The connectors wrapping the filaments at their ends prevent the coiled filaments from opening uncontrollably before discharge and act as barriers that need to be overcome first for the shaft to be everted.

Reviewer #3 (Remarks to the Author):

This is an exemplary study of nematocyst structure and operation with state-of-the-art live-cell imaging, confocal and electron microscopy, and stunning image analysis. The accompanying videos are superb, allowing the viewer to grasp the organization of this marvel of evolutionary engineering. The clips showing the image reconstruction of the nematocyst structure from serial sections and the videos of shaft/tubule extension are remarkable. Wow, yes, I am very pleased to see this as a reviewer. Technically, the paper seems very strong to me, but the accompanying text begins with a significant error and I have a couple of additional points for consideration.

Reviewer 3 Point 1: Abstract: The nematocyst is described both as an organelle and a cell. It is a cell that contains a harpoon made from modified secretory organelles. The discharge mechanism is a form of explosive exocytosis.

Response 3 Point 1: We thank the reviewer for their comments regarding the designation of the nematocyte (specialized cell type) and the nematocyst (explosive organelle). To address this point, we edited the abstract and the introduction in lines 12-14 and lines 28-30.

Reviewer 3 Point 2: Introduction, line 33. This is an unforgivable error. Nematocyst discharge is the fastest biological movement: discharge of the Hydra nematocyst can be completed in less than a microsecond. Nüchter et al. estimated 700 nanoseconds. Tubule eversion is much slower and can take as long as a millisecond, but this secondary process is not very swift at all compared with the mechanisms of fungal spore discharge, pollen ejection, and so on. What makes the nematocyst so awe-inspiring is that initial nanosecond eruption. This should be said very clearly in the Introduction even though the new experiments concern the slower processes of shaft and tubule extension.

Response 3 Point 2: We added a paragraph in the introduction describing the results reported previously by Holstein & Tardent (1984) on the kinetics of nematocysts discharge in Hydra including the total combined duration of the extremely fast discharge phase and the following slower tubule elongation phase (see lines 32-33). The estimated nanosecond initial discharge kinetics reported by Nüchter et al. is described on lines 33-36. We added a sentence to indicate that this initial phase of discharge is comparable to other ejectile processes found in nature such as fungal spore discharge, pollen ejection and the action of the ballistic projectiles of dinoflagellates (lines 36-38). We also indicated that the second process of tubule or thread eversion is a much slower process (lines 45-47).

Reviewer 3 Point 3: Introduction. The description of the shaft and tubule eversion would benefit from a diagram right up front. L.51 refers to shaft eversion; l.53-4 states that tubule eversion

differs from shaft eversion “as the tubule everts simply by turning inside-out.” Eversion means to turn inside-out and if the shaft does this, readers may wonder why the tubule is different? I understand that the mechanics of Phase II shaft eversion and the single phase of tubule eversion are very different, but this may not be clear as explained on l.51-

Response 3 Point 3: To address this concern, we rewrote the section in the introduction to describe the tubule eversion as the evagination of a tubular structure without the triple helical arrangement of filaments on lines 65-66. While we considered creating a new diagram, we ultimately felt it better to keep the diagram at the end in Figure 4.

Reviewer 3 Point 4: The mechanisms are clarified later, on l.153- when the authors distinguish between the Phase I process of shaft discharge without any eversion and the Phase II process of eversion. Eversion of a portion of the shaft then proceeds to eversion of the connecting tubule. Maybe Phase I and Phase II of shaft extension should be introduced in the sentence on l.47-49: “. . . piercing the target (Phase I), and later everts (Phase II) to form a lumen . . .”

Response 3 Point 4: To address this point we have now included a sentence to describe these two distinct phases of operation (piercing the target and forming the lumen) on lines 50-51.

Otherwise, this is a fantastic piece of work.

Thank you very much.

a

Reviewer Figure 1 | The snapshots of nematocysts captured during discharge
The snapshot of discharging nematocysts captured on tentacle tips in TRITC treated animals.

Reviewer Figure 2 | SEM image of discharge nematicysts
Arrows indicate the everted shaft filaments exhibiting a drill-bit like structure.

Reviewer Figure 3| Discharged threads stained labeled anti-VASA or anti-Nco14 antibodies

a. Image of threads after antibody staining with Rabbit anti-VASA antibody (white) and TRITC (green). a', Magnified view of antibody-stained tubule showing Rabbit anti-VASA (white) and TRITC (green) co-staining. b, Image of threads after antibody staining with Rabbit anti-Nco14 antibody (white) and TRITC (green) labeled thread. b', Magnified view of antibody-stained tubule showing Rabbit anti-Nco14(white) and TRITC labeled thread (green).

Primary polyp tentacles

Reviewer Fig. 4 | Effects of *v1g243188* gene knockdown on the discharged thread structures on primary polyp tentacles.

The structure of the shaft and the tubule after knockdown with shRNAs in TRITC labeled, discharged threads. **a**, The structure of the shaft and the tubule after knockdown with scramble control shRNA. **b-d**, The structure of the shaft and the tubule after knockdown with *v1g243188* shRNA.

Scramble shRNA

v1g243188

Reviewer Fig.5 | Effects of *v1g243188* gene knockdown on the thread structures. The structure of the shaft and the tubule after knockdown with shRNAs in TRITC labeled, discharged threads on tentacles of the primary polyps. Magnified regions of the tubule (dashed boxes) were shown in insets. **a**, The structure of the shaft and the tubule after knockdown with scramble control shRNA. **Upper Left panel** Magnified regions of the tubule were shown. **b**, The structure of the shaft and the tubule after knockdown with *v1g243188* shRNA. **Lower left panel:** Magnified regions of the tubule were shown. Scale bar, 1um

Reviewer Fig.6: Knockdown ejected structures appear to be less straight than their control counterparts in spinalin (*v1g243188*). Control (N=10) and knockdown (N=14) ejected structures were tracked point by point manually and unit directional vectors were calculated for each segment. Autocorrelation functions were calculated for vector dot products, averaged over all structures and plotted with error bars.

Reviewer Fig. 7 | Discussion of the data figure provided by Reviewer 2. a, The structure of an unevverted nematocyst shaft (likely a p-mastigophore) detached from the capsule without eversion. Apical flaps, V-notch, dart and capsule-shaft connector was shown. **b**, Fluorescence image of a TRITC (green) labeled, undischarged but burst capsule, pre-treated with

Chondroitinase ABC. **Left panel:** TRITC signal from the undischarged shaft and the tubule without eversion. **Middle panel:** Corresponding WGA (magenta) labeled capsule and thread structures. **Right panel:** Combined TRITC (green) and WGA (magenta) channels showing the expanded shaft (dashed line). The end of two uncoiled shaft filaments, the expanded capsule-shaft and shaft-tubule connectors, the partially digested wall of the capsule and the tubule was shown. Scale bar 1um

Reviewer Fig. 8 | TRITC and WGA co-staining of discharged capsules

a, Fluorescence image of a discharged capsule stained with WGA (magenta) and TRITC (green) after Chondroitinase ABC treatment. The triple arrangement of the apical flaps and the TRITC enriched spot (arrow) were shown. **b**, The image of the nematocyst capsule seen in WGA (magenta) channel, showing the enrichment of the WGA in the capsule outer wall and its borders (magenta, double arrows). **c**, The image of the nematocyst capsule seen in TRITC (green channel, showing the enrichment of the TRITC in the capsule inner wall and its borders (green, dashed arrows). The arrow indicates TRITC (green) enriched dot. Scale bar 1 μm .

REVIEWERS' COMMENTS

Reviewer #2 (Remarks to the Author):

The authors addressed all my concerns in an adequate manner. The manuscript is ready for acceptance. I congratulate them for a nice piece of work.

Minor comments:

l. 121: who noted the tip of the shaft is formed by staggered lamellae converging at (a) small area

l. 254: of TRITC intensity indicated that a fraction of the dye was also incorporated (into) the shaft structure